# Ca²⁺ entry through Na$_V$ channels generates submillisecond axonal Ca²⁺ signaling

Naomi AK Hanemaaijer[1,2†], Marko A Popovic[1†‡], Xante Wilders[1], Sara Grasman[1], Oriol Pavón Arocas[1§], Maarten HP Kole[1,2*]

[1]Department of Axonal Signaling, Netherlands Institute for Neuroscience (NIN), Royal Netherlands Academy of Arts and Sciences (KNAW), Amsterdam, Netherlands; [2]Cell Biology, Neurobiology and Biophysics, Department of Biology, Faculty of Science, Utrecht University, Utrecht, Netherlands

**\*For correspondence:**
m.kole@nin.knaw.nl

[†]These authors contributed equally to this work

**Present address:** [‡]Molecular Cell Biology and Immunology, Amsterdam UMC, Location VUmc, Amsterdam, Netherlands; [§]Sainsbury Wellcome Centre for Neural Circuits and Behaviour, University College London, London, United Kingdom

**Competing interests:** The authors declare that no competing interests exist.

**Abstract** Calcium ions (Ca²⁺) are essential for many cellular signaling mechanisms and enter the cytosol mostly through voltage-gated calcium channels. Here, using high-speed Ca²⁺ imaging up to 20 kHz in the rat layer five pyramidal neuron axon we found that activity-dependent intracellular calcium concentration ($[Ca^{2+}]_i$) in the axonal initial segment was only partially dependent on voltage-gated calcium channels. Instead, $[Ca^{2+}]_i$ changes were sensitive to the specific voltage-gated sodium (Na$_V$) channel blocker tetrodotoxin. Consistent with the conjecture that Ca²⁺ enters through the Na$_V$ channel pore, the optically resolved $I_{Ca}$ in the axon initial segment overlapped with the activation kinetics of Na$_V$ channels and heterologous expression of Na$_V$1.2 in HEK-293 cells revealed a tetrodotoxin-sensitive $[Ca^{2+}]_i$ rise. Finally, computational simulations predicted that axonal $[Ca^{2+}]_i$ transients reflect a 0.4% Ca²⁺ conductivity of Na$_V$ channels. The findings indicate that Ca²⁺ permeation through Na$_V$ channels provides a submillisecond rapid entry route in Na$_V$-enriched domains of mammalian axons.

## Introduction

Ca²⁺ ions crossing the neuronal plasma membrane are critically involved in depolarization and distribute in the cytosol in spatial microdomains and organelles to regulate a wide range of processes ranging from gene expression to fast transmitter release (*Berridge, 2006*; *Neher and Sakaba, 2008*). In axons, voltage-gated Ca²⁺ (Ca$_V$) channels at presynaptic terminals open in response to a single action potential (AP), raising intracellular Ca²⁺ concentrations ($[Ca^{2+}]_i$) in nanodomains from ~50 nM up to ~10 μM to increase transmitter vesicle release rates by the power of ~4 (*Helmchen et al., 1997*; *Schneggenburger and Neher, 2000*). In response to APs, large and local $[Ca^{2+}]_i$ transients are typically also observed in the axon initial segment (AIS) and nodes of Ranvier (*Callewaert et al., 1996*; *Bender and Trussell, 2009*; *Yu et al., 2010*; *Gründemann and Clark, 2015*; *Zhang and David, 2016*; *Clarkson et al., 2017*). At these sites, Ca²⁺ currents have been implicated in AP initiation and propagation by a local depolarizing action of the inward current or by activating the large conductance, Ca²⁺- and voltage-dependent K⁺ (BK$_{Ca}$) channels modulating burst firing probability and limiting frequency-dependent AP failure rates (*Bender and Trussell, 2009*; *Yu et al., 2010*; *Hirono et al., 2015*). The Ca$_V$ channel subtypes identified in axons are both cell type- and species-dependent and include the T-, P/Q- or N-type Ca$_V$ channels (*Callewaert et al., 1996*; *Bender and Trussell, 2009*; *Yu et al., 2010*; *Gründemann and Clark, 2015*; *Zhang and David, 2016*). At the AIS in particular the T-type Ca²⁺ channel mediates AP-dependent Ca²⁺ influx (*Bender and Trussell, 2009*; *Martinello et al., 2015*; *Fukaya et al., 2018*; *Jin et al., 2019*). However, in the prefrontal cortical pyramidal neuron AIS about 70% of the AP-

**eLife digest** Nerve cells communicate using tiny electrical impulses called action potentials. Special proteins termed ion channels produce these electric signals by allowing specific charged particles, or ions, to pass in or out of cells across its membrane. When a nerve cell 'fires' an action potential, specific ion channels briefly open to let in a surge of positively charged ions which electrify the cell. Action potentials begin in the same place in each nerve cell, at an area called the axon initial segment. The large number of sodium channels at this site kick-start the influx of positively charged sodium ions ensuring that every action potential starts from the same place.

Previous research has shown that, when action potentials begin, the concentration of calcium ions at the axon initial segment also increases, but it was not clear which ion channels were responsible for this entry of calcium. Channels that are selective for calcium ions are the prime candidates for this process. However, research in squid nerve cells gave rise to an unexpected idea by suggesting that sodium channels may not exclusively let in sodium but also allow some calcium ions to pass through. Hanemaaijer, Popovic et al. therefore wanted to test the routes that calcium ions take and see whether the sodium channels in mammalian nerve cells are also permeable to calcium.

Experiments using fluorescent dyes to track the concentration of calcium in rat and human nerve cells showed that calcium ions accumulated at the axon initial segment when action potentials fired. Most of this increase in calcium could be stopped by treating the neurons with a toxin that prevents sodium channels from opening. Electrical manipulations of the cells revealed that, in this context, the calcium ions were effectively behaving like sodium ions. Human kidney cells were then engineered to produce the sodium channel protein. This confirmed that calcium and sodium ions were indeed both passing through the same channel.

These results shed new light on the relationship between calcium ions and sodium channels within the mammalian nervous system and that this interplay occurs at the axon initial segment of the cell. Genetic mutations that 'nudge' sodium channels towards favoring calcium entry are also found in patients with autism spectrum disorders, and so this new finding may contribute to our understanding of these conditions.

evoked $[Ca^{2+}]_i$ remains following pharmacological block of T-type $Ca_V$ channels (*Clarkson et al., 2017*). Furthermore, evidence for a clustering of T-type $Ca_V$ channels at the AIS is ambiguous and immunofluorescence or immuno-gold labeling studies show a density which is comparable to soma-todendritic or spine compartments (*McKay et al., 2006*; *Martinello et al., 2015*).

Several other mechanisms may contribute to axoplasmic $[Ca^{2+}]_i$ elevations in the AIS. Firstly, $Ca^{2+}$ levels could rise due to $Ca^{2+}$-induced $Ca^{2+}$ release from intracellular sources such as the endoplasmic reticulum (ER). Most AISs contain ER cisternae organelles consisting of stacks of membranes expressing the store-operated ryanodine receptor (RyR), inositol 1,4,5-triphosphate receptor 1 ($IP_3R1$) and sarcoplasmic ER $Ca^{2+}$ ATPase (SERCA) pumps (*Benedeczky et al., 1994*; *King et al., 2014*; *Antón-Fernández et al., 2015*). The coupling of transmembrane $Ca^{2+}$ entry with intracellular store release may generate a local activity-dependent rise of $[Ca^{2+}]_i$. However, a contribution of ER stores to AIS $[Ca^{2+}]_i$ remains to be directly demonstrated. Secondly, near the peak of the AP the electrogenic $Na^+$-$Ca^{2+}$ exchanger (NCX) reverses direction and imports $Ca^{2+}$. A reverse mode of operation has not only been implicated in pathological $[Ca^{2+}]_i$ elevations in axons during hypoxia and injury (*Stys et al., 1991*; *Iwata et al., 2004*), but also occurs during trains of APs in nodes and neighboring internodes (*Zhang and David, 2016*). Finally, one alternative pathway that has yet to be directly examined in mammalian cortical axons involves the voltage-gated $Na^+$ ($Na_V$) channels. Studies in the squid giant axon combining electrophysiological recordings with $Ca^{2+}$ imaging showed that an early component of depolarization-induced $Ca^{2+}$ entry is tetrodotoxin (TTX)-sensitive (*Baker et al., 1971*; *Meves and Vogel, 1973*; *Brown et al., 1975*). Voltage-clamp recordings from axons and perfusing distinct ionic solutions provided a quantitative estimate that $Na_V$ channels may pass divalent $Ca^{2+}$ ions with permeability ratios ($P_{Ca}/P_{Na}$) up to 0.10 (*Hille, 1972*; *Meves and Vogel, 1973*). $Ca^{2+}$ permeability of $Na_V$ channels has also been shown in cardiac cells and hippocampal neurons (*Akaike and Takahashi, 1992*; *Aggarwal et al., 1997*; *Santana et al., 1998*) but whether this extends to the cortical axons remains to be examined.

Here, using wide-field $Ca^{2+}$ imaging with a high-speed CCD camera enabling detection of $[Ca^{2+}]_i$ changes at high sensitivity and high temporal resolution (*Jaafari et al., 2014*; *Ait Ouares et al., 2016*), we explored the various pathways of $Ca^{2+}$ entry in axons of rat thick-tufted neocortical layer 5 (L5) pyramidal neurons. We found that during subthreshold depolarizations $[Ca^{2+}]_i$ transients were highly compartmentalized to the AIS and nodes of Ranvier. While these transients were amplified by ER store release, the trigger was only modestly accounted for by $Ca_V$ channels. The largest fraction of activity-dependent $[Ca^{2+}]_i$ was TTX-sensitive and overlapped with the rapid gating of $Na_V$ channels. Experiments in HEK-293 cells transfected with the human $Na_V1.2$ channel confirmed that TTX-sensitive $Na^+$ currents were sufficient to generate $[Ca^{2+}]_i$ elevations. Together, the data suggest that $[Ca^{2+}]_i$ dynamics in the mammalian AIS are predominantly mediated by a rapid $Ca^{2+}$ entry through $Na_V$ channels.

## Results

### Activity-dependent compartmentalized $Ca^{2+}$ entry in layer five axons

Thick-tufted L5 pyramidal neurons, also called L5B or pyramidal tract neurons, are the largest pyramidal neurons in the cortex and integrate synaptic inputs from all cortical layers, playing a central role in cognitive tasks including perception (*Groh et al., 2010*; *Ramaswamy and Markram, 2015*; *Takahashi et al., 2016*). Their large axons (~1.5 μm in diameter) send long-range output projections to the thalamus, striatum and spinal cord, but within the cortex branch sparsely and have a trajectory perpendicular to the pia providing an excellent anatomical arrangement to image and record from. To optically record the spatial profile of axonal $[Ca^{2+}]_i$ we made somatic whole-cell patch-clamp recordings from neurons filled with the high-affinity $Ca^{2+}$ indicator Oregon Green BAPTA 1 (OGB-1, 100 μM) and imaged epifluorescence signals along the proximal region of the main axon (*Figure 1*). We first used subthreshold depolarizations evoked by artificial excitatory postsynaptic potentials (a-EPSPs, 100 Hz, peak depolarization 17.0 ± 0.6 mV, *n* = 15; *Figure 1a*). Examination of the spatial profile revealed that $Ca^{2+}$ signals were observed in the AIS and hot spots separated with regular distances along the axon (locations 2, 4 and 6; *Figure 1a*). In order to examine whether the $[Ca^{2+}]_i$ hot spots corresponded to nodes of Ranvier, we post-hoc stained for βIV-spectrin and biocytin, and found indeed overlap between subthreshold $[Ca^{2+}]_i$ rise and spectrin-enriched sites (*Figure 1b*). In the same cells we examined the spatial profile of $\Delta[Ca^{2+}]_i$ in response to a single AP evoked with a brief square current injection (*Figure 1c*). As expected from back- and forward-propagating APs with much higher depolarizations (~100 mV), large $[Ca^{2+}]_i$ transients were observed widespread throughout all axonal and somatodendritic domains. Population analysis showed that AP-induced $[Ca^{2+}]_i$ transients were similar between AIS and nodes (one-way ANOVA followed by Tukey's multiple comparison test, p<0.0001, differences between all groups were significant (p<0.05) except between AIS and node (p=0.13) and between internode (IN) and dendrites (p=0.85); *Figure 1c,d*). Interestingly, also during a-EPSPs the $[Ca^{2+}]_i$ transients in the AIS and the first nodes were highly comparable, while $[Ca^{2+}]_i$ signals in the internodal and dendritic domains were an order of magnitude smaller (one-way ANOVA with Tukey's multiple comparison test, p<0.0001, differences between all groups were significant (p<0.0001), except between AIS and node (p=0.38) and IN and Dend (p=0.97); *Figure 1a,d*). Similar experiments in L5 neocortical pyramidal neurons in slices from human temporal cortex also revealed a-EPSP evoked $\Delta[Ca^{2+}]_i$ in the AIS, but not in the dendrite, suggesting that subthreshold sensitive $[Ca^{2+}]_i$ transients are conserved across mammalian species (*Figure 1—figure supplement 1*). Together, these results show that activity-dependent $[Ca^{2+}]_i$ transients are spatiotemporally compartmentalized and $Ca^{2+}$ entry dynamics are similar in the axoplasm of the AIS and nodes.

### Giant saccular organelle amplifies activity-dependent $[Ca^{2+}]_i$ transients in the AIS

The thick-tufted L5 pyramidal neuron AIS contains a unique variant of cisternal organelle characterized by a continuous tubular organization of smooth ER, called the giant saccular organelle (*Antón-Fernández et al., 2015*). Cisternal organelles with smooth ER express synaptopodin (synpo), RyR, the $IP_3$ receptor 1, and SERCA that are thought to contribute to $Ca^{2+}$ release, buffering and storage (*Bas Orth et al., 2007*; *King et al., 2014*). We hypothesized that these organelles could generate

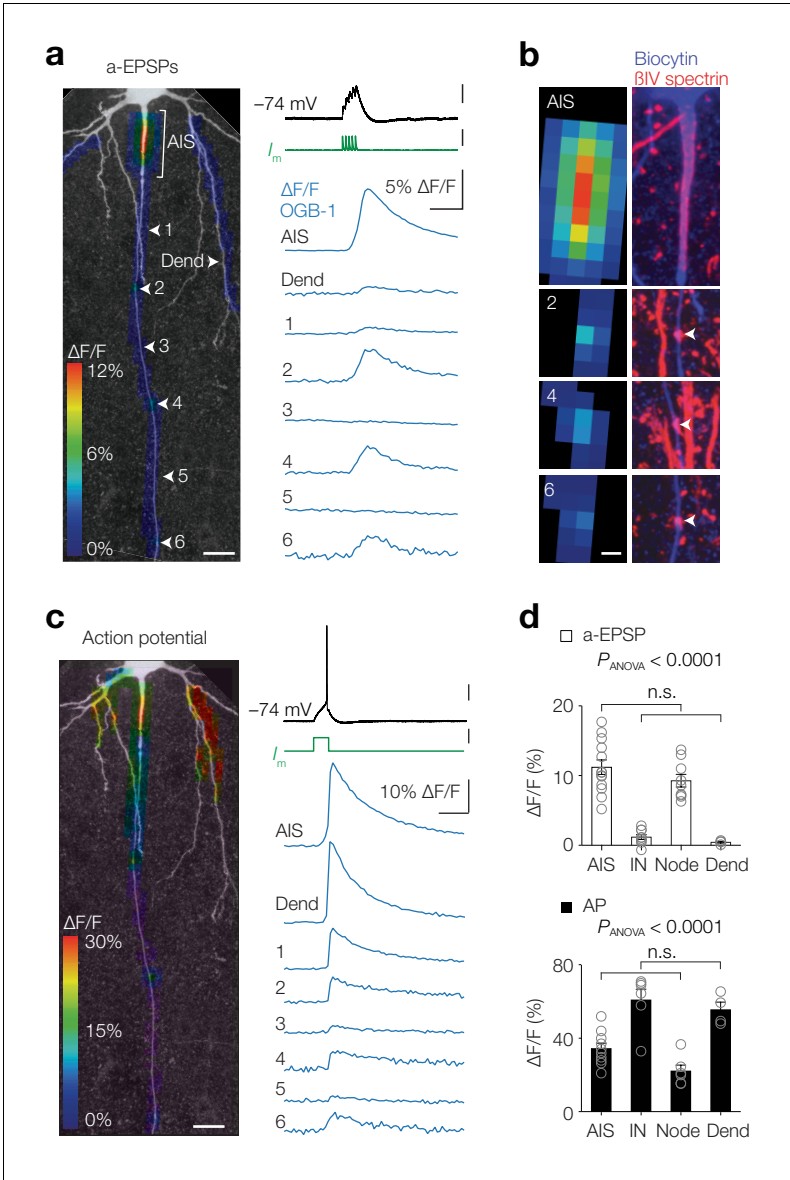

**Figure 1.** Activity-dependent compartmentalized Ca$^{2+}$ entry in layer five axons. (**a**) *Left*, Color-coded maximal ΔF/F of OGB-1 imaged in a L5 pyramidal neuron axon in response to an a-EPSP (five subthreshold current injections at 100 Hz) overlaid with a *z*-projection of biocytin-streptavidin (grey) of the same neuron. White arrowheads indicate regions of interest from which example traces are shown *right*. Scale bar, 20 μm. *Right*, ΔF/F traces from locations specified *left*. For illustrative purposes, ΔF/F traces represent averages of ~400 trials. Top to bottom scale bars, 10 mV, 1 nA, 5% ΔF/F, 100 ms. (**b**) *Left*, higher magnification of the regions of interest shown in a. *Right*, maximal *z*-projection of biocytin-streptavidin (blue) and ßIV spectrin (red). White arrows indicate nodes or Ranvier. Sites with a-EPSP-evoked [Ca$^{2+}$]$_i$ transients were all positive to ßIV spectrin (*n* = 15 AISs and 23 nodes from *n* = 15 axons). Scale bar, 5 μm. (**c**) Same axon and locations as in a with color-coded maximal ΔF/F in response to an AP. Scale bars, 20 mV, 0.5 nA, 10% ΔF/F, 100 ms and 20 μm. (**d**) *Top*, Population data for peak ΔF/F in response to a-EPSPs in the AIS (*n* = 13), first internode (IN, *n* = 9), the first Node (*n* = 9) and basal dendrite (Dend, *n* = 4), one-way ANOVA with Tukey's multiple comparisons test, p<0.0001, *Bottom*, peak ΔF/F in response to a single AP in the AIS (*n* = 10), first internode (IN, *n* = 6), the first Node (*n* = 6) and basal dendrite (Dend, *n* = 4), one-way ANOVA with Tukey's multiple comparisons test, p<0.0001. Open circles represent individual cells and bars show the population mean ± s.e.m. Data available in *Figure 1—source data 1*. See also *Figure 1—figure supplement 1*.

The online version of this article includes the following source data and figure supplement(s) for figure 1:

**Source data 1.** Activity-dependent compartmentalized Ca$^{2+}$ entry in layer five axons.

*Figure 1 continued on next page*

*Figure 1 continued*

**Figure supplement 1.** EPSP-evoked evoked $\Delta[Ca^{2+}]_i$ in human layer five pyramidal neuron AIS.

Ca$^{2+}$-induced Ca$^{2+}$-release, thereby contributing to domain-selective activity-dependent $[Ca^{2+}]_i$ transients (*Figure 1*). Triple immunostaining for synpo, Ankyrin G and biocytin confirmed that the cisternal organelle was present along the entire axis of the AIS and spatially overlapped with the subthreshold-evoked $[Ca^{2+}]_i$ transients ($n$ = 19; *Figure 2a,b*). However, while subthreshold depolarization-induced Ca$^{2+}$ transients were present in the nodes, synaptopodin expression was not detected ($n$ = 10 nodes; *Figures 1* and *2a*). To experimentally test whether AIS Ca$^{2+}$-store release contributes to activity-dependent $[Ca^{2+}]_i$ transients we performed experiments with standard intracellular solution and subsequently re-patched the same cell with a solution containing ryanodine (200 µM, blocking RyR-mediated Ca$^{2+}$ release) and heparin (5 mg/ml, competitively inhibiting IP$_3$-evoked Ca$^{2+}$ release; *Figure 2c*). Blocking Ca$^{2+}$ release significantly lowered $\Delta F/F$ Ca$^{2+}$ peak transients in the AIS, both for the subthreshold- and AP-evoked $[Ca^{2+}]_i$ changes (a-EPSP, 53.2% reduction, p=0.006; AP, 34.3% reduction, p=0.02, one-tailed ratio paired t-tests, $n$ = 5; *Figure 2d*). Consistent with the AIS-specific location of the giant saccular organelle, store blockers had no effect on AP-evoked $\Delta[Ca^{2+}]_i$ in the basal dendrite (p=0.48, $n$ = 3; *Figure 2d*). Furthermore, since the stores contribute significantly to AIS Ca$^{2+}$ levels, blocking store release could act as a low-pass filter for Ca$^{2+}$ level kinetics, reducing rise and decay times. However, blocking Ca$^{2+}$-store release did not alter the rise- or decay time kinetics in the AIS ($\tau_{act}$, p=0.52; $\tau_{de-act}$, p=0.18, two-tailed paired *t*-tests, $n$ = 5; *Figure 2e*). These data suggest that the giant saccular organelle amplifies activity-dependent $[Ca^{2+}]_i$ changes selectively in the AIS.

## Ca$_V$ channels and NCX have a limited role in activity-dependent Ca$^{2+}$ entry at the AIS

Ca$^{2+}$ release from internal stores is likely triggered by Ca$^{2+}$ entry via neuronal voltage-dependent plasmalemmal routes. To test whether AIS $[Ca^{2+}]_i$ changes require Ca$^{2+}$ from the extracellular space, we bath applied 2.5 mM of the Ca$^{2+}$ chelator EGTA which effectively lowered the extracellular Ca$^{2+}$ concentration ($[Ca^{2+}]_o$) from 2 mM to ~437 nM, thereby reducing the driving force for Ca$^{2+}$ (see Materials and methods). Ca$^{2+}$ imaging at the AIS (OGB-1, 100 µM) showed that EGTA almost fully abolished the subthreshold-evoked $\Delta[Ca^{2+}]_i$ (90.7% reduction, one-tailed ratio paired t-test, p=0.0031, $n$ = 4; *Figure 3a,b*). Similarly, the AP-generated $\Delta[Ca^{2+}]_i$ was almost extinguished after bath application of EGTA (92.8% reduction, p=0.0011, $n$ = 4; *Figure 3a,b*).

Next, we hypothesized that the transmembrane pathway for Ca$^{2+}$ entry in the AIS during subthreshold stimuli could be explained by the low-voltage gated Ca$_V$ channels (T- and R-type). However, bath application of the highly selective T-type (Ca$_V$3.1–3) blocker TTA-P2 (1 µM, *Choe et al., 2011*) or nickel (Ni$^{2+}$, 100 µM) did not significantly reduce Ca$^{2+}$ signals (one-tailed ratio paired *t*-tests; TTA-P2, p=0.17, $n$ = 4; Ni$^{2+}$, p=0.063, $n$ = 5; *Figure 3b,c*). We next blocked R-type Ca$_V$ channels, by puffing SNX-482 (500 nM) locally to the AIS, but this did not lead to a reduction in subthreshold $[Ca^{2+}]_i$ rise either (SNX-482, p=0.11, $n$ = 3). Furthermore, consistent with their more depolarized voltage range of activation, the L-type channels did not affect subthreshold $\Delta[Ca^{2+}]_i$ (20 µM isradipine, p=0.14; 10 µM nimodipine, p=0.41, both $n$ = 4; *Figure 3b*) and the block of N-type and P/Q-type channels, by local application of ω-conotoxin MVIIC (2 µM) to the AIS, also failed to reduce Ca$^{2+}$ signals (p=0.42, $n$ = 5; *Figure 3b*). Furthermore, a combined block of T- and L-type channels did not affect the peak $\Delta F/F$ in the AIS (TTA-P2 and isradipine, p=0.12, $n$ = 3; *Figure 3b*).

Although application of the T-type blockers TTA-P2 and Ni$^{2+}$ was ineffective to block subthreshold $[Ca^{2+}]_i$ rise, in the same neurons it did reduce the peak $\Delta F/F$ evoked by a single AP by almost 20% (TTA-P2, 18.7%, p=0.021, $n$ = 4; Ni$^{2+}$, 19.7% block, p=0.013, $n$ = 5; *Figure 3b,c*). In addition, isradipine reduced the AP-evoked $\Delta[Ca^{2+}]_i$ in the AIS by 14.9% (isradipine, p=0.0070, $n$ = 4) and the alternative L-type blocker nimodipine showed a non-significant blocking trend (nimodipine, p=0.060, $n$ = 4; *Figure 3b*). A combined application of T- and L-type channel blockers (1 µM TTA-P2 and 20 µM isradipine) caused a 27.2% reduction of peak $\Delta F/F$, showing a sublinear summation of two blocking agents (TTA-P2 and isradipine, p=0.0071, $n$ = 5; *Figure 3b*). In contrast, the R-type Ca$_V$ channel blocker SNX-482 (500 nM) did not reduce the AP-evoked $\Delta[Ca^{2+}]_i$ (SNX-482, p=0.29, $n$ = 3;

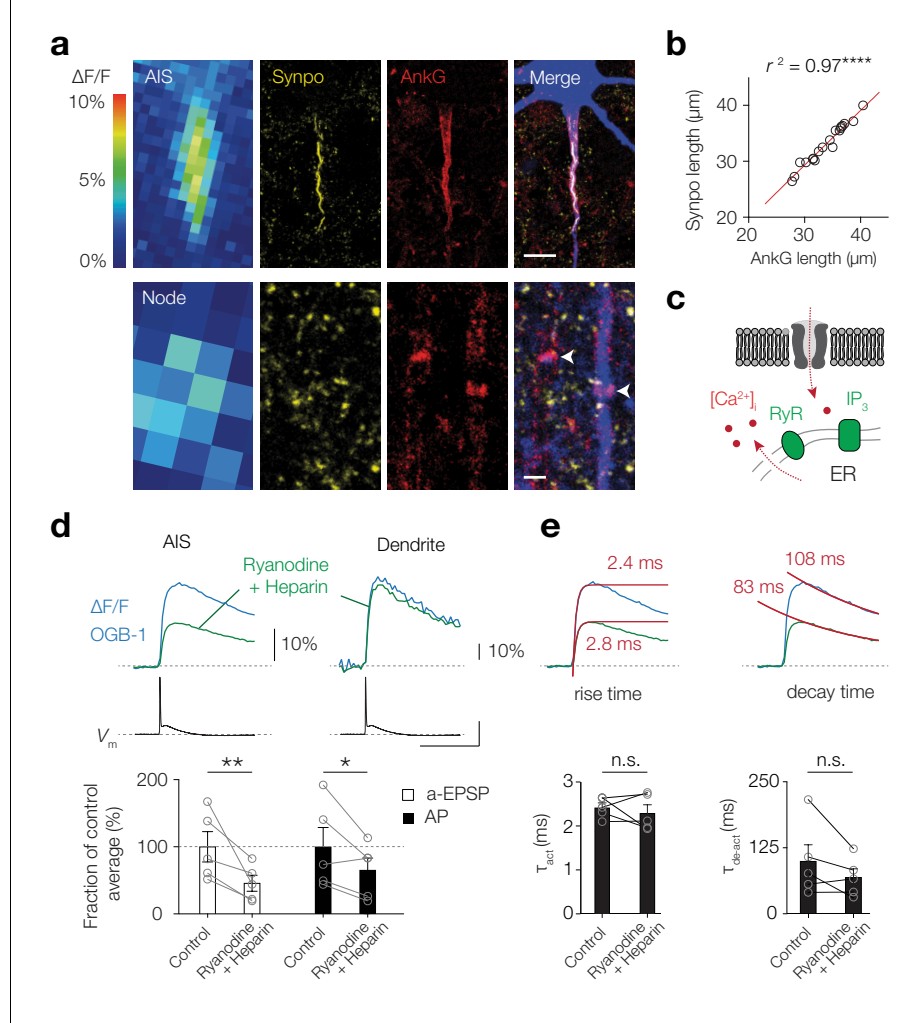

**Figure 2.** Giant saccular organelle amplifies activity-dependent $\Delta[Ca^{2+}]_i$ in the AIS. (**a**) Example color-coded maximal ΔF/F profile (100 μM OGB-1) in response to a-EPSPs in the AIS (*top*) and Node (*bottom*) compared to *z*-projections for synaptopodin (Synpo, yellow), Ankyrin-G (AnkG, red) and biocytin/streptavidin (blue) of the same axon. White arrows indicate the locations of nodes, in the imaged axon (blue) and a neighboring one, both without synaptopodin. The many small Synpo positive puncta are likely co-localized with subclasses of spines (*Benedeczky et al., 1994*; *Bas Orth et al., 2007*). Scale bars, 10 μm and 1 μm. (**b**) The length of synaptopodin fluorescence linearly scales with Ankyrin G length. Red trace, linear regression fit with $y = 1.005 x - 0.95$, $r^2 = 0.966$, ****$p<0.0001$, $n = 19$. (**c**) Schematic of $Ca^{2+}$-induced $Ca^{2+}$ release by internal ER stores. (**d**) *Top*, example ΔF/F (OGB-1) transients from AIS (left) and dendrite (right) in response to an AP ($V_m$, black) in control conditions (blue) and re-patched with blockers (green). Scale bars, 10% ΔF/F, 50 mV and 50 ms. *Bottom*, population data of the peak ΔF/F in the AIS in response to a-EPSP (white bars) and AP (black bars) before and after store release block, one-tailed ratio paired t-tests, **$p=0.0060$, *$p=0.021$, $n = 5$. Open circles and connecting lines represent paired recordings from individual cells and bars show the population mean ± s.e.m. (**e**) *Top*, $Ca^{2+}$ transients fitted with a single exponential function (red) to the rise (left) and decay time (right) in response to an AP, red number indicates the τ. *Bottom*, comparison of the rise and decay time (two-tailed paired *t*-tests, $\tau_{act}$, $p=0.52$, $\tau_{de\text{-}act}$, $p=0.18$, $n = 5$). Open circles and connecting lines represent paired recordings from individual cells and bars show the population mean ± s.e.m. Data available in *Figure 2—source data 1*.

The online version of this article includes the following source data for figure 2:

**Source data 1.** Giant saccular organelle amplifies activity-dependent $\Delta[Ca^{2+}]_i$ in the AIS.

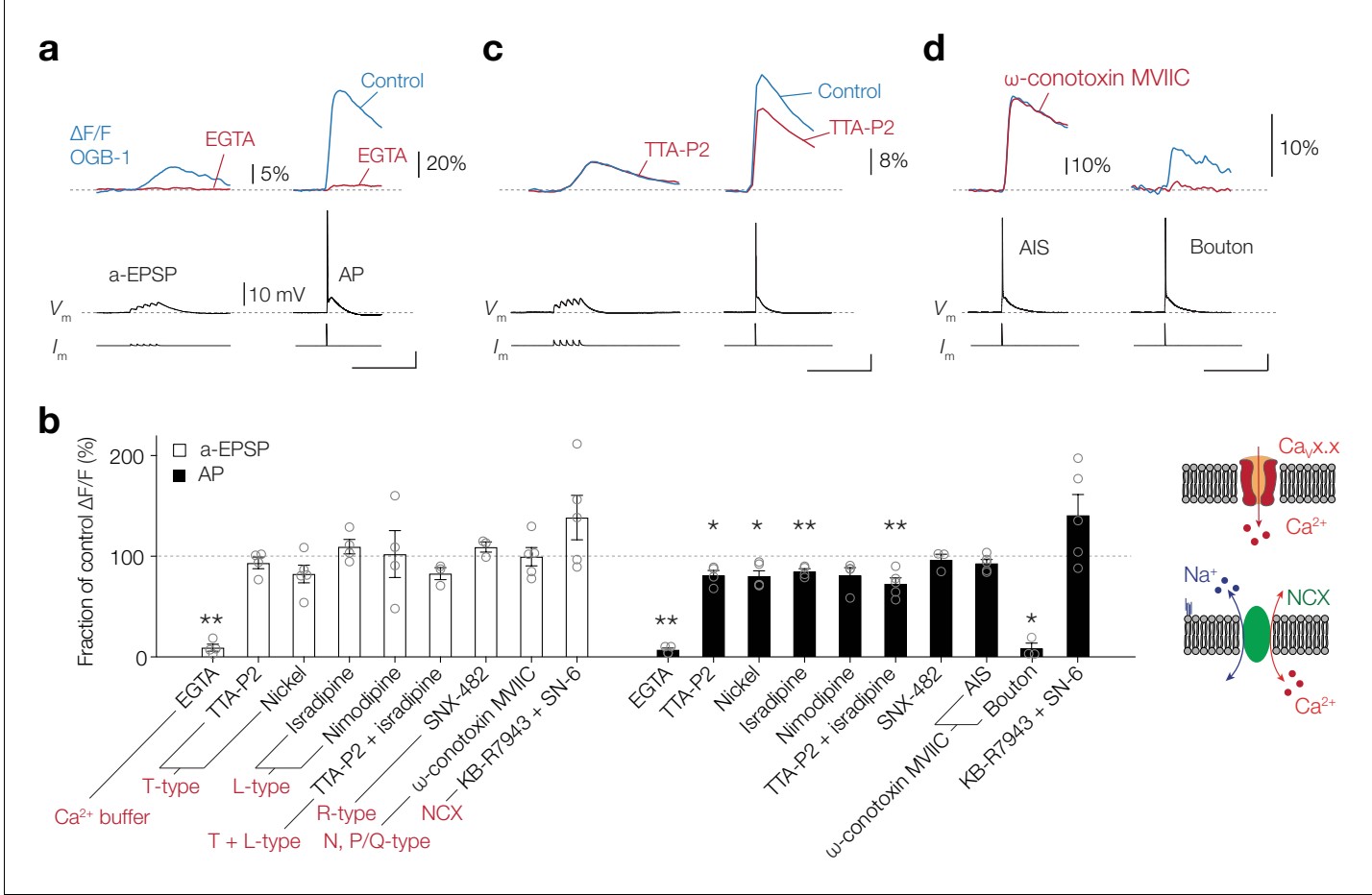

**Figure 3.** $Ca_V$ channels play a partial role in activity-dependent $Ca^{2+}$ entry at the AIS. (**a**) *Top*, example traces of ΔF/F (100 μM OGB-1) in the AIS evoked by an a-EPSP (left) and AP (right), before (blue) and after (red) bath application of EGTA (2.5 mM). *Bottom*, somatic $V_m$ and current-clamp protocols. Scale bars bottom right, 20 mV, 100 ms. (**b**) *Left*, population data for the effect of an extracellular $Ca^{2+}$ buffer, $Ca_V$ channel blockers and NCX on the peak $\Delta[Ca^{2+}]_i$ signal in the AIS in response to an a-EPSP (open bars) and AP stimulation (closed bars). Data are shown as ratio to the pre-drug peak $Ca^{2+}$ responses measured in the same neuron. One-tailed ratio paired t-tests, *p<0.05 and **p<0.01. Open circles represent individual cells and the bars show the mean ± s.e.m. *Right*, schematics of $Ca^{2+}$ entry in the axoplasm via $Ca_V$ channels (*top*) or NCX (*bottom*). (**c**) *Top*, example traces of ΔF/F in the AIS evoked by an a-EPSP (left) and AP (right), before (blue) and after (red) bath application of TTA-P2 (1 μM). *Bottom*, somatic $V_m$ and current-clamp protocols. Scale bar, 20 mV, 100 ms. (**d**) *Top*, example traces of ΔF/F evoked by an AP in the AIS (left) and presynaptic bouton of the same neuron (right), before (blue) and after (red) puff application of ω-conotoxin MVIIC (2 μM). *Bottom*, somatic $V_m$ and current-clamp protocols. Scale bar, 20 mV, 100 ms. Data available in *Figure 3—source data 1*.

The online version of this article includes the following source data for figure 3:

**Source data 1.** $Ca_V$ channels play a partial role in activity-dependent $Ca^{2+}$ entry at the AIS.

*Figure 3b*). Local application of the ω-conotoxin MVIIC (2 μM) showed a non-significant trend to block the peak ΔF/F (6.8%, p=0.064, *n* = 5; *Figure 3b*). As a positive control experiment, we imaged a collateral bouton of the same neuron and used local application of ω-conotoxin MVIIC which almost completely abolished the peak ΔF/F by 91.5%, consistent with the presence of N- and P/Q-type $Ca_V$ channel subtypes in presynaptic terminals (p=0.021, *n* = 3; *Figure 3b*). Finally, given the unexpected remaining $[Ca^{2+}]_i$ transients in the AIS in the presence of $Ca_V$ channel blockers, we hypothesized that NCX may contribute to activity-dependent $[Ca^{2+}]_i$ increase in the AIS. At the resting membrane potential NCX exports $Ca^{2+}$ from the cytoplasm to maintain $[Ca^{2+}]_i$ near ~50 nM, however $Na^+$ entry may promote instantaneous $Ca^{2+}$ influx by a reverse mode of operation (*Yu and Choi, 1997*; *Stys and LoPachin, 1997*; *Figure 3b*). To examine its contribution, we pharmacologically blocked NCX by combined bath application of KB-R7943 (20 μM) and SN-6 (10 μM). The results

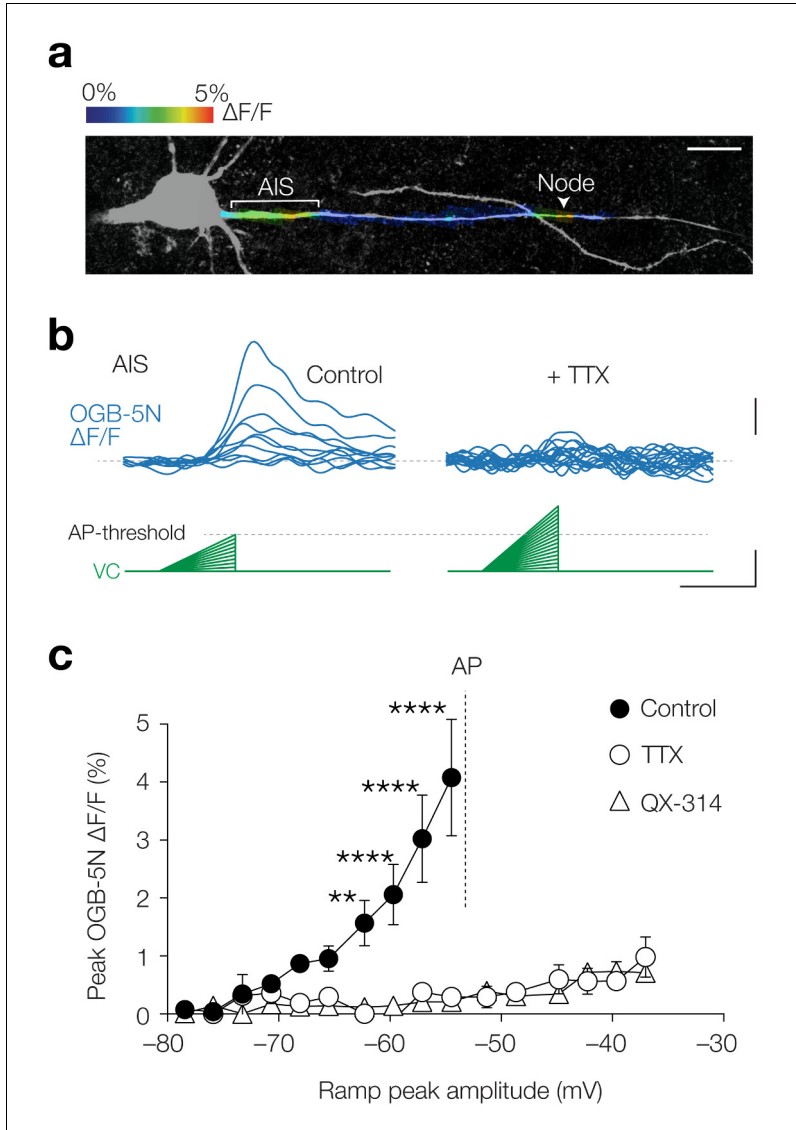

**Figure 4.** Subthreshold $[Ca^{2+}]_i$ changes are TTX sensitive. (**a**) Example image of color-coded maximal $\Delta F/F$ (1 mM OGB-5N) along the proximal axon in response to a voltage ramp from –78.5 mV to maximally –54.5 mV. Acquisition rate was 125 Hz. The color-coded image is overlaid with a maximal *z*-projection of a confocal scan of biocytin/streptavidin staining (grey) of the same axon. Note the compartmentalized $Ca^{2+}$ influx at the AIS and node (indicated by a white arrow). Scale bar, 20 μm. (**b**) Example traces of OGB-5N $\Delta F/F$ reveal increasing $Ca^{2+}$ responses to voltage ramps with increasing incline in control (*left*) but not in TTX (1 μM, *right*). Traces filtered with one-pass binomial (three point). Scale bars, 1% $\Delta F/F$, 10 mV and 50 ms. (**c**) Population data for maximal OGB-5N $\Delta F/F$ versus voltage ramp peaks in control (closed circles, *n* = 4), in TTX (open circles, *n* = 4) or QX-314 (open triangles, 6 mM, *n* = 4). Two-way ANOVA with Tukey's multiple comparisons test, **p<0.01, ****p<0.0001. AP threshold is indicated with a gray line. Data are shown as mean ± s.e.m. Data available in *Figure 4—source data 1*.

The online version of this article includes the following source data for figure 4:

**Source data 1.** Subthreshold $[Ca^{2+}]_i$ changes are TTX sensitive.

---

showed, however, no change in the subthreshold nor AP-evoked $\Delta[Ca^{2+}]_i$ (two-tailed ratio paired t-tests, a-EPSP, p=0.16; AP, p=0.13, *n* = 5, respectively; *Figure 3b*).

In summary, these data show that while transmembrane $Ca^{2+}$ influx from the extracellular space is required for activity-evoked $\Delta[Ca^{2+}]_i$, none of the $Ca_V$ channels played a role in the subthreshold depolarization, whereas T- and L-type $Ca_V$ channels partially contributed to the AP-evoked influx.

## Subthreshold- and AP-evoked Ca$^{2+}$ entry at the AIS requires TTX-sensitive channels

What could be the source of the remaining component of Ca$^{2+}$ influx at the AIS? Both in hippocampal neurons and heart muscle cells, Ca$^{2+}$ currents have been described which are not blocked by Ni$^{2+}$ nor by other known Ca$_V$ channel antagonists, but instead are sensitive to the highly selective Na$_V$ channel blocker (TTX), and therefore called $I_{Ca(TTX)}$ (*Akaike and Takahashi, 1992*; *Aggarwal et al., 1997*). In Na$^{+}$-free extracellular solution $I_{Ca(TTX)}$ resembles the Na$^{+}$ current and activates at potentials as negative as –70 mV while peaking at –30 mV (*Akaike and Takahashi, 1992*). To examine the presence of $I_{Ca(TTX)}$ in L5 axons we took advantage of the low-affinity indicator Oregon Green BAPTA 5N (OGB-5N, 1 mM; *Figure 4a*), which gives smaller fluorescent signals but is linear over a wider range of [Ca$^{2+}$]$_i$ compared to OGB-1 ($K_d$ 20 µM vs. 170 nM, respectively). We used the voltage-clamp configuration and injected depolarizing ramps of 50 ms with increasing slopes (from 0.0 to 0.55 mV ms$^{-1}$) with a maximum peak at ~95% of the AP threshold (*Figure 4b*), thereby studying the same depolarization range and duration as the a-EPSPs used in *Figures 1–3*. The results showed that Ca$^{2+}$ influx was strongly compartmentalized to the AIS and nodal axolemma (*Figure 4a*) consistent with the a-EPSP evoked OGB-1 transients (*Figure 1*). Remarkably, bath application of TTX almost completely abolished [Ca$^{2+}$]$_i$ elevations, even at depolarizations above the AP threshold (at –54.5 mV, 92.8% block, Cohen's *d*: 1.88, two-way ANOVA, p<0.0001, *n* = 4; *Figure 4b,c*). As an alternative to TTX we used the quaternary amine Na$_V$ channel inhibitor QX-314, which plugs the open state of the Na$_V$ channel from the internal side. Similar to TTX, with 6 mM QX-314 added to the pipette solution voltage ramps did not evoke Ca$^{2+}$ transients (at –54.5 mV, 94.8% block, Cohen's *d*: 1.92, control vs. QX-314, p<0.0001, *n* = 4, TTX vs. QX-314, p=0.97, *n* = 4; *Figure 4c*). Although QX-314 at this concentration has been reported to also block voltage-gated Ca$^{2+}$ currents (*Talbot and Sayer, 1996*), subthreshold-evoked [Ca$^{2+}$]$_i$ was not mediated by Ca$_V$ channels (*Figure 3b*). The near complete block by two distinct Na$_V$ channel blockers therefore indicates an important role of Na$_V$ channels in mediating subthreshold axonal Ca$^{2+}$ influx.

We next investigated whether Na$_V$ channels also contribute to AP-evoked Δ[Ca$^{2+}$]$_i$ (*Figure 5a*). To dissociate a putative role of Na$_V$ channels to pass Ca$^{2+}$ ions from generating the AP depolarization of ~100 mV we first imaged Ca$^{2+}$ at the AIS in current-clamp, subsequently applied 1 µM TTX and imaged Ca$^{2+}$ transients evoked by the recorded AP waveform injected as a voltage command ('AP-clamp'). The results showed a near complete abolishment of Δ[Ca$^{2+}$]$_i$ in the presence of TTX (one-way ANOVA with Tukey's multiple comparisons test, CC vs. VC, 89.5% reduction, p<0.0001, *n* = 7; *Figure 5a,b*). However, this [Ca$^{2+}$]$_i$ peak amplitude reduction could also be due to an incomplete voltage- and space-clamp of the AIS for fast voltage transients. The small diameter of the axon has a high axial resistance, acting as a low-pass filter for the antidromic AIS action potentials (*Hamada et al., 2016*) which also may attenuate the orthodromic voltage spread into the axon. To examine the possibility that axonal APs are attenuated in the somatic AP clamp configuration, we optically recorded JPW3028, a fast fluorescent voltage indicator that remains stable over long recording periods and is highly linear over a large voltage range (~250 mV, *Figure 5—figure supplement 1*). Consistent with the voltage loss, when we injected the AP-clamp in the soma in the presence of TTX and optically recorded $V_m$ in the AIS, we observed a significant ~2 fold reduction in the AP peak amplitude (one-way ANOVA with Tukey's multiple comparisons test, VC vs. CC, p=0.014; *Figure 5a,c*). To restore peak depolarization in the presence of TTX and reliably compare the Ca$^{2+}$ transients evoked by equal depolarization, we doubled the amplitude of the somatic AP-clamp (VC ×2). With this protocol both the peak depolarization and AP half-width in the AIS were indistinguishable from the control APs (peak JPW, VC ×2 vs. CC, one-way ANOVA with Tukey's multiple comparisons test, p=0.75, *n* = 4, *Figure 5c*, half-width in JPW, VC ×2 vs. CC, one-way ANOVA with Tukey's multiple comparisons test, p=0.36, *n* = 4, not shown). TTX blocked 65.5% of the AP-evoked Δ[Ca$^{2+}$]$_i$ (peak OGB-5N, Cohen's *d*: 5.49, VC ×2 vs. CC, p<0.0001, *n* = 8; *Figure 5b*). Taken together, these data suggest that a large fraction of both subthreshold-depolarization and AP-evoked Ca$^{2+}$ ions enter the axoplasm through TTX-sensitive channels at the AIS.

## Axonal [Ca$^{2+}$]$_i$ follows gating kinetics of Na$_V$ channels

Whether $I_{Ca(TTX)}$ is carried by a specific TTX-sensitive Ca$_V$ channel or reflects Ca$^{2+}$ permeating directly through the Na$_V$ channel remains debated (*Santana et al., 1998*; *Cruz et al., 1999*;

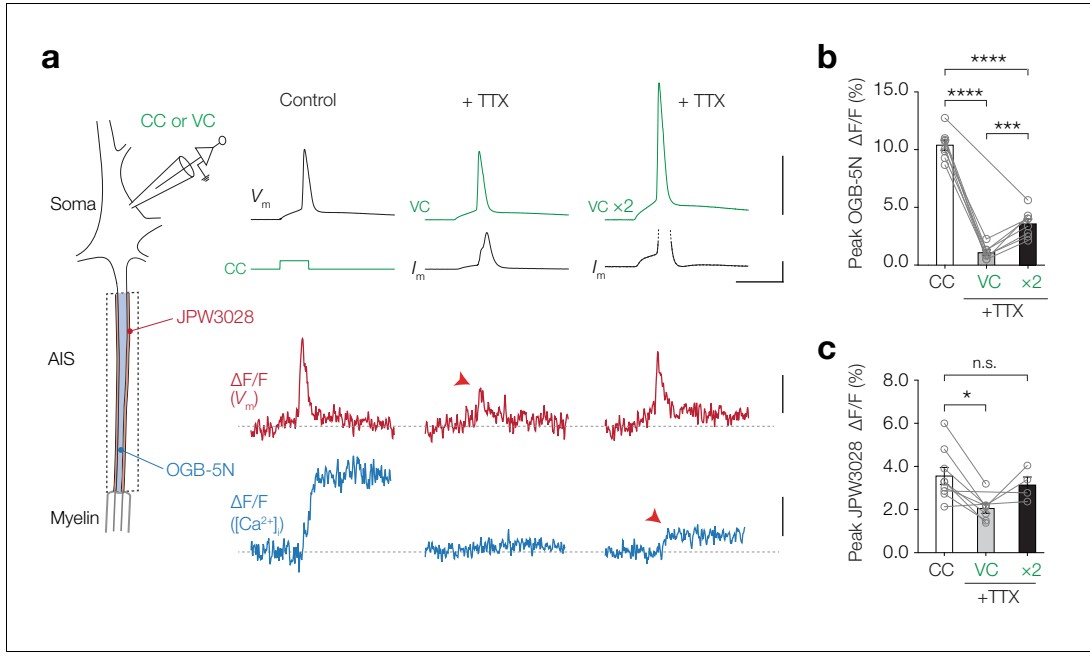

**Figure 5.** AP-evoked $[Ca^{2+}]_i$ changes are TTX sensitive. (a) *Left*, schematic of the experimental setup; electrophysiological recording from the soma and fluorescence recording from AIS. *Right*, example traces from electrophysiology (recorded traces in black, applied commands in green), ΔF/F JPW3028 (red) for voltage imaging and ΔF/F OGB-5N (blue) for $Ca^{2+}$ imaging. Left panel shows example traces from a current clamp recording, middle panel was performed in AP-clamp in the presence of TTX (1 μM) and right panel was performed in AP-clamp scaled two-fold (VC ×2) in the presence of TTX. The JPW3028 and OGB-5N experiments were performed in separate cells. Scale bars from top to bottom, 100 mV, 5 nA, 5 ms, 2% ΔF/F and 5% ΔF/F. (b) Peak OGB-5N ΔF/F in response to an AP ($n = 8$), AP clamp + TTX (VC, $n = 7$) and double AP clamp + TTX (VC ×2, $n = 8$). One-way ANOVA with Tukey's multiple comparisons test, ***p=0.0005, ****p<0.0001, $n = 8$. Circles and connecting lines represent paired recordings in individual cells and bars show the population mean ± s.e.m. (c) Peak JPW3028 ΔF/F in response to an AP ($n = 9$), AP clamp + TTX (VC, $n = 7$) and double AP clamp + TTX (VC ×2, $n = 4$). One-way ANOVA with Tukey's multiple comparisons test, *p=0.014. Circles and connecting lines represent paired recordings in individual cells and bars show the population mean ± s.e.m. Data available in *Figure 5—source data 1*. See also *Figure 5—figure supplement 1*.

The online version of this article includes the following source data and figure supplement(s) for figure 5:

**Source data 1.** AP-evoked $[Ca^{2+}]_i$ changes are TTX sensitive.

**Figure supplement 1.** JPW3028 fluorescence linearly increases with membrane potential over a wide range of voltages.

---

*Heubach et al., 2000*; *Chen-Izu et al., 2001*; *Guatimosim et al., 2001*). We hypothesized that if $Ca^{2+}$ ions enter the AIS cytoplasm by flowing through $Na_V$ channels, $I_{Ca(TTX)}$ should reflect the time course of $I_{Na}$. To measure submillisecond rapid events with fluorescence in the small axon (diameter ~1.5 μm), we optimized multiple imaging parameters enabling the acquisition of fluorescence at 20 kHz (see Materials and methods). Using a $Na^+$-sensitive indicator (sodium-binding benzofuran isophthalate, SBFI, 1 mM) in combination with OGB-5N (1 mM) showed that the two indicators were indistinguishable in their rising phase during an AP, suggesting that $Ca^{2+}$ entry at the AIS may be as rapid as $Na^+$ entry (*Figure 6—figure supplement 1*).

In order to quantify the kinetics of $I_{Na}$ and $I_{Ca}$ more directly, we next used a voltage-clamp approach. Near ~20°C and –40 mV, $Na_V$ channels open at least one order of magnitude faster compared to the T-type $Ca_V$ channels (~200 μs [*Schmidt-Hieber and Bischofberger, 2010*] vs. ~5 ms [*Perez-Reyes et al., 1998*], respectively) which may be sufficiently different to compare against the kinetics of optically recorded $[Ca^{2+}]_i$ at the AIS. To determine the specific activation kinetics of $I_{Na}$ and T-type $I_{Ca}$ ($I_{CaT}$) in L5 pyramidal neurons we measured total inward current ($I_{Na} + I_{Ca}$) by depolarizing the soma with a step to –35 mV and pharmacologically isolated $Na^+$ and $Ca^{2+}$ current components by bath application of 1 μM TTX or 100 μM $Ni^{2+}$, respectively (*Figure 6a*). The activation time

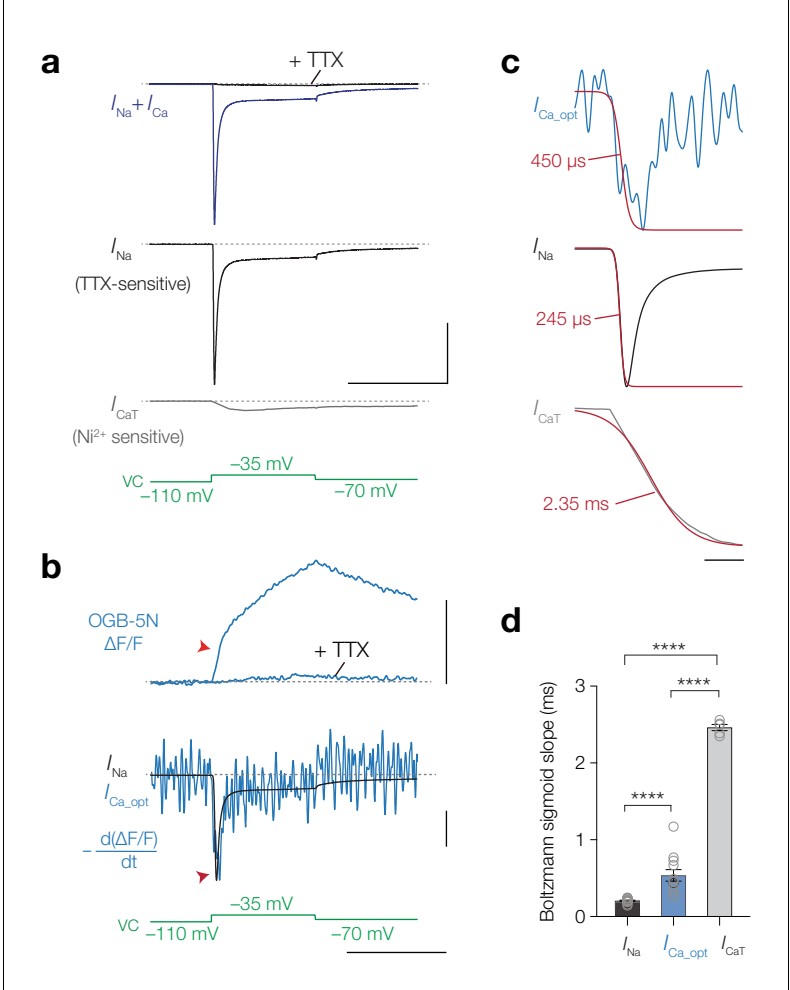

**Figure 6.** Temporal derivative of AIS $[Ca^{2+}]_i$ mirrors the opening kinetics of $Na_V$ channels. (**a**) *Top to bottom*, total inward current in response to a voltage-clamp step to –35 mV (blue) and after application of 1 µM TTX (black), $I_{Na}$; the difference between the total current and the TTX-sensitive current, $I_{CaT}$, obtained after application of 100 µM $Ni^{2+}$, voltage clamp protocol (green). Scale bars, 10 nA and 50 ms. (**b**) *Top to bottom*: ΔF/F OGB-5N of an AIS imaged at 20 kHz before and after application of 1 µM TTX, the electrically recorded somatic $I_{Na}$ of the same neuron overlaid with the first temporal derivative of ΔF/F ($-d(ΔF/F)\,dt^{-1}$) representing $I_{Ca\_opt}$, voltage clamp protocol. Optically and electrically recorded traces filtered with a 3-point binomial filter for 100 iterations. Red arrows indicate the rapid component visible in the ΔF/F and in the overlaid $I_{Na}$ and $I_{Ca\_opt}$. Scale bars, 20% ΔF/F, 10%$^{-1}$/6.3 nA and 50 ms. (**c**) *Top to bottom*: $I_{Ca\_opt}$, $I_{Na}$ and $I_{CaT}$ fit with a Boltzmann sigmoid function (red). Slope values are indicated. Scale bar, 5 ms. (**d**) Population data for the slope values for $I_{Na}$ (*n* = 17), $I_{Ca\_opt}$ (*n* = 10) and $I_{CaT}$ (*n* = 5). One-way ANOVA with Tukey's multiple comparisons test, ****p<0.0001. Open circles represent individual cells, bars indicate the population mean ± s.e.m. Data available in *Figure 6—source data 1*. See also *Figure 6—figure supplement 1*.

The online version of this article includes the following source data and figure supplement(s) for figure 6:

**Source data 1.** Temporal derivative of AIS $[Ca^{2+}]_i$ mirrors the opening kinetics of $Na_V$ channels.
**Figure supplement 1.** AP-evoked $Ca^{2+}$ and $Na^+$ entry follow identical kinetics.

constant of the total inward current was identical to $I_{Na}$ (single exponential fit $\tau_{total}$ = 438.2 µs vs. $\tau_{Na}$ = 440.3 µs, one-way ANOVA with Tukey's multiple comparison test, p>0.999, *n* = 6), whereas the total current was substantially faster in comparison to $I_{CaT}$ ($\tau_{total}$ = 438.2 µs vs. $\tau_{CaT}$ = 4.8 ms, p<0.0001, *n* = 5; *Figure 6a*). The initial fraction of the inward current was thus primarily generated by $I_{Na}$. The large difference in gating could provide a temporal window to distinguish $Ca^{2+}$ entry via $Na_V$ channels or T-type $Ca_V$ channels. Theoretical and experimental work show that low-affinity $Ca^{2+}$ indicators, like OGB-5N, are capable of tracking rapidly activating $Ca^{2+}$ currents when imaged at

high speed: the first time derivative of ΔF/F (dΔF/F dt$^{-1}$) overlaps with electrically recorded $I_{Ca}$, providing a mean to optically resolve the time course of $I_{Ca}$ (*Sabatini and Regehr, 1998*; *Jaafari et al., 2014*; *Ait Ouares et al., 2016*).

Imaging OGB-5N (1 mM) at 20 kHz in the AIS we observed that ΔF/F comprised of two separate time courses, a fast initial rise followed by a slower rising phase (*Figure 6b*). Both components were almost completely abolished by TTX, leaving only a small transient reflecting putatively $I_{CaT}$ ($n = 6$). We quantitatively compared the activation time constants of $I_{Na}$, $I_{Ca\_opt}$ (dΔF/F dt$^{-1}$) and $I_{CaT}$ by resampling the electrically recorded $I_{Na}$ and $I_{CaT}$ to 20 kHz and filtering both electrical and optical traces identically (see Materials and methods). Multiple hallmarks of $I_{Na}$ matched with $I_{Ca\_opt}$: both traces showed a rapid inward component, followed by inactivation and a persistent component (*Figure 6b,c*). In comparison, $I_{CaT}$ lacked both the rapid activation and inactivation time constants (*Figure 6a,c*). Given the lower signal-to-noise ratio in the optical traces we fitted Boltzmann sigmoid functions to the rising phase to compare the slopes of the optically and electrically recorded currents (*Figure 6c*). The average slope of $I_{Ca\_opt}$ was significantly faster compared to the activation of $I_{CaT}$ (~500 μs vs. ~2.5 ms, respectively, one-way ANOVA with Tukey's multiple comparison test, p<0.0001) and slower compared to $I_{Na}$ (~500 μs vs. ~200 μs, $I_{Ca\_opt}$ vs. $I_{Na}$p<0.0001; *Figure 6c,d*). The small difference between $I_{Na}$ and $I_{Ca\_opt}$ may be explained by the equilibration time of OGB-5N (~200 μs) (*Ait Ouares et al., 2016*), local differences between Na$_V$ channels in the soma and AIS or the presence of Ca$^{2+}$-store release in the AIS. Together, the findings indicate that the current mediating [Ca$^{2+}$]$_i$ at the AIS resembles Na$_V$ channel kinetics.

## Calcium influx through Na$_V$1.2 channels

The results suggest that Ca$^{2+}$ ions could enter the cytoplasm by permeation through the Na$_V$ channel pore. Previous studies showed Ca$^{2+}$ influx through the cardiac Na$_V$1.5 channel (*Cruz et al., 1999*; *Guatimosim et al., 2001*). To examine whether Na$_V$ channel isoforms of the axon initial segment also enable Ca$^{2+}$ influx we performed experiments in HEK-293 cells which were transfected with the human gene *SCN2A* encoding Na$_V$1.2 channel with auxiliary β1 and β2 subunits and EGFP tag (Materials and methods; *Figure 7a*). Whole-cell recording revealed Na$^+$ currents in EGFP$^+$ cells but not in non-transfected cells (average peak current density –115.7 ± 28.4 pA/pF, $n = 10$ vs. –3.7 ± 1.9 pA/pF at –20 mV, $n = 5$, respectively; *Figure 7b*). The inward currents were completely abolished by 1 μM TTX (96.1 ± 1.3%, $n = 7$, one-tailed Wilcoxon matched-pairs signed rank test, p=0.0078, *Figure 7c*) and the voltage-dependence of activation and inactivation revealed midpoints at –25.4 ± 2.1 mV and –74.1 ± 3.9 mV, respectively ($n = 10$, *Figure 7d,e*), consistent with previous work (*Ben-Shalom et al., 2017*), indicating a highly selective expression of Na$_V$1.2 channels. Next, we filled the transfected cells with 100 μM OGB-1 and imaged the fluorescence changes in response to a train of depolarizing pulses (200 Hz for 1 s, –120 to –30 mV steps; *Figure 7f*). We observed an increase in ΔF/F in every EGFP$^+$ cell, indicating an influx of Ca$^{2+}$ (average peak 0.46 ± 0.18% ΔF/F, range: 0.06–1.4% ΔF/F, $n = 7$; *Figure 7f–h*). To test whether the [Ca$^{2+}$]$_i$ increase required Na$_V$ channel opening we bath applied TTX (1 μM), revealing a significant decrease in the peak ΔF/F (92.1 ± 3.8% reduction, one-tailed Mann Whitney test, p=0.012, $n = 4$; *Figure 7h*). The results indicate that molecular expression and opening of Na$_V$1.2 channels suffices to mediate transmembrane Ca$^{2+}$ influx.

## Estimating Ca$^{2+}$ conductivity of Na$_V$ channels with computational modeling

Our findings are in agreement with the depolarization-induced Ca$^{2+}$ entry in the squid axon which is tetrodotoxin (TTX)-sensitive and reflected a 1% conductivity of Na$_V$ channels for Ca$^{2+}$ ions (*Baker et al., 1971*). To estimate the conductivity ratios ($g_{Ca}/g_{Na}$) in L5 axons we performed computational simulations. Ca$^{2+}$ entry through Na$_V$ channels was implemented by adding an ohmic Ca$^{2+}$ ion mechanism into a mathematical 8-state Na$_V$ channel model that calculated the current carried by Ca$^{2+}$ ($I_{Ca(Na)}$) and Na$^+$ ($I_{Na}$) (see Materials and methods). A single compartment containing $I_{Na}$ and $I_{Ca(Na)}$ together with high voltage-gated and T type-gated Ca$_V$ channel models ($I_{CaH}$ and $I_{CaT}$, respectively) showed that with an axonal AP waveform $I_{Ca(Na)}$ is activated during the first microseconds of AP onset, rapidly inactivates and is temporally separated from $I_{CaH}$ and $I_{CaT}$ (*Figure 8—figure supplement 1*). Next, to estimate the $g_{Ca}/g_{Na}$ we made a multicompartmental model of a L5 pyramidal

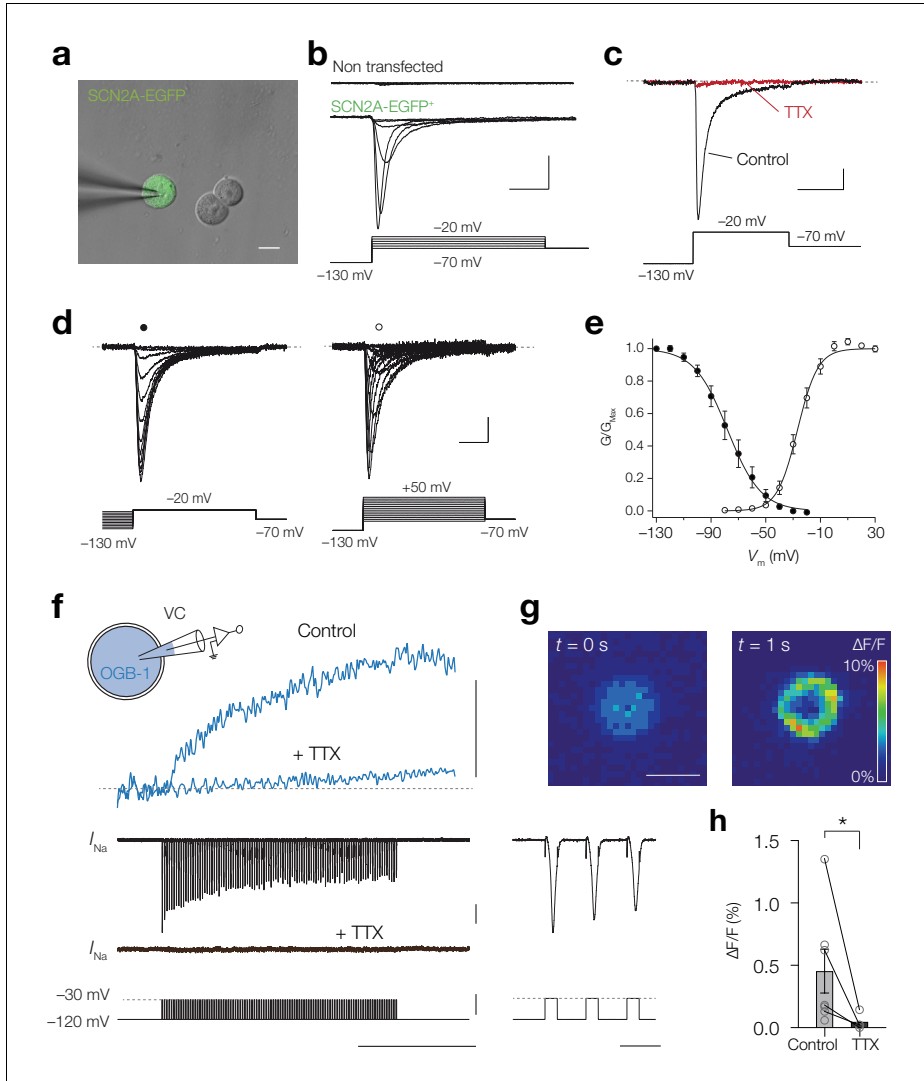

**Figure 7.** Na$_V$1.2 channels mediate a Ca$^{2+}$ influx. (a) Combined brightfield and fluorescence image of a whole-cell recording from a SCN2A-EGFP$^+$ HEK-293 cell (green). Scalebar, 10 μm. (b) Example traces of $I_{Na}$ recorded in response to depolarizing voltage command potentials (*bottom*) for a non-transfected (*top*) and SCN2A-EGFP$^+$ cell (*middle*), scale bars indicate 0.5 nA and 5 ms. (c) Peak $I_{Na}$ traces before (black) and after TTX application (red) in response to a depolarizing voltage step (*bottom*), scale bars indicate 100 pA and 10 ms. (d) Example current traces of steady-state inactivation (*left*) and activation protocols (*right*), scale bars indicated 100 pA and 5 ms. (e) Population data for steady-state activation and inactivation curves, circles and error bars indicate mean ± s.e.m. Lines represent Boltzmann fits to the mean data. (f) Schematic of the experiment: a SCN2A-EGFP$^+$ HEK-293 cell was recorded in voltage-clamp and filled with OGB-1 (0.1 mM) after which a train of depolarizing pules (–120 to –30 mV, 200 Hz, 1 s) was applied, *top to bottom*, ΔF/F before and after bath application of 1 μM TTX (blue), the recorded currents and voltage command potentials, *right*, magnification of the first three action currents, scale bars from top to bottom represent 1% ΔF/F, 0.5 nA, 100 mV, 500 ms and 10 ms. (g) Color-coded average ΔF/F of 100 frames before onset (*left*) and at the end (*right*) of the voltage command, scale bar indicates 10 μm. (h) OGB-1 ΔF/F is significantly higher in control (*n* = 7) than after application of TTX (*n* = 4), one-tailed Mann Whitney test, p=0.0121. Data available in *Figure 7—source data 1*.

The online version of this article includes the following source data for figure 7:

**Source data 1.** Na$_V$1.2 channels mediate a Ca$^{2+}$ influx.

neuron (*Figure 8a*, including detailed reconstructions of the AIS and nodal domains (see Figure 2a within *Hamada et al., 2016*). Based on multiple experimentally recorded parameters we constrained the model AP and found that a peak conductance density of Na$_V$ channels of 16,000 and 850 pS µm$^{-2}$ in the AIS and soma, respectively, reproduced the recorded AP and matched with AP-evoked $\Delta[Na^+]_i$ imaged in the AIS (see Materials and methods and *Figure 8a*). Subsequently, $[Ca^{2+}]_i$ was simulated based on mathematical equations representing $Ca^{2+}$ diffusion and extrusion, endogenous stationary $Ca^{2+}$ buffers (taken together as $\kappa_s$) and was supplemented with the buffering capacities of the specific $Ca^{2+}$ indicators (*Fink et al., 2000*) (see Materials and methods). The $Ca^{2+}$ extrusion threshold and rates were adjusted to approximate the experimentally imaged peak and decay time course of measured OGB-5N $\Delta F/F$ in the AIS (*Figure 8—figure supplement 2*). To determine the absolute rise in $[Ca^{2+}]_i$ produced exclusively by Na$_V$ channels, we performed additional experiments in which we imaged $[Ca^{2+}]_i$ while blocking Ca$_V$ channels that contributed to AP-evoked $\Delta[Ca^{2+}]_i$: T- and L-type calcium channels (TTA-P2 and isradipine, respectively, see *Figure 3b*). Using calibrated ratiometric bis-Fura-2 (200 µM) imaging, we found that during 1 AP, $Ca^{2+}$ entry though Na$_V$ channels induces a peak $\Delta[Ca^{2+}]_i$ of 55.6 nM (n = 4; *Figure 8b*, *Figure 8—figure supplement 2*). Since ~35% of AP-evoked $\Delta[Ca^{2+}]_i$ is caused by internal store amplification (*Figure 2d*) ~36 nM is mediated by transmembrane $Ca^{2+}$ entry via Na$_V$ channels (*Figure 8b*). We subsequently simulated these experiments in the multicompartmental model by removing Ca$_V$ channels and including the buffering properties of 200 µM bis-Fura-2. Varying endogenous buffering ($\kappa_s$) between 1 and 100 we updated $g_{Ca}/g_{Na}$ to obtain a 36 nM rise of free $[Ca^{2+}]_i$ at the AIS. A $\kappa_s$ of ~100 corresponds to dendritic buffering

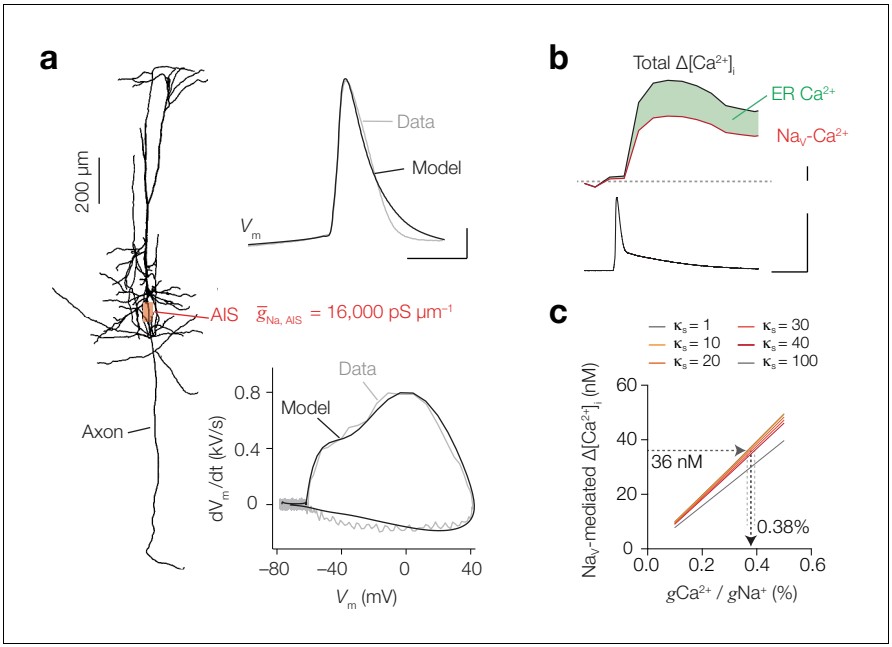

**Figure 8.** Computational simulation of $Ca^{2+}$ conductivity by Na$_V$ channels predicts a conductivity ratio of 0.38%. (a) *Left*, morphology of the conductance-based multi-compartmental model. *Right*, Na$_V$ channel density in the AIS was estimated by optimizing to $\Delta[Na^+]_i$, $V_m$ (*top*) and phase-plane plot (*bottom*) in simulation (black) to the experimental data (gray). Scale bars, 200 µm, 20 mV and 0.5 ms. (b) Example trace of calibrated ratiometric imaging of bis-Fura2 to measure absolute changes in $\Delta[Ca^{2+}]_i$ in response to a single AP (bottom), the experiment was performed in the presence of T- and L-type CaV blockers, so $\Delta[Ca^{2+}]_i$ is mediated by Na$_V$ channels (red) and amplified by internal store release (35%, green), scale bars indicate 10 nM, 100 mV and 10 ms. (c) Dependence of Na$_V$-mediated peak AP $\Delta[Ca^{2+}]_i$ on conductance ratio ($g_{Ca}/g_{Na}$) for varying endogenous buffer capacities ($\kappa_s$ = 1–100). $\Delta[Ca^{2+}]_i$ was measured and modeled in a cell with 200 µM bis-Fura-2 present and T- and L-type Ca$_V$ channels blocked. See also *Figure 8—figure supplement 1* and *Figure 8—figure supplement 2*.

The online version of this article includes the following figure supplement(s) for figure 8:

**Figure supplement 1.** $I_{Ca}$ compared to $I_{Ca(Na)}$ in a single compartment model.

**Figure supplement 2.** Calibration and modeling of $[Ca^{2+}]_i$.

capacities (*Cornelisse et al., 2007*), while axonal buffering capacities are reported to be lower (10–40, *Klingauf and Neher, 1997*; *Jackson and Redman, 2003*; *Delvendahl et al., 2015*). When changing $\kappa_s$ between 10–40 the $g_{Ca}/g_{Na}$ ratio was 0.38% ($\kappa_s$ = 10: 0.37%, $\kappa_s$ = 40: 0.39%; *Figure 8c*).

## Spatiotemporal distribution of Ca²⁺ entry routes under physiological conditions

Using the 0.38% conductivity ratio we next evaluated how Ca²⁺ currents through Na_V and Ca_V channels spatiotemporally varied across the neuronal compartments without the buffering capacities of externally applied Ca²⁺ dyes (*Figure 9a,b* and *Figure 8—figure supplement 2*). The simulations showed that the Δ[Ca²⁺]ᵢ from one AP reached a peak concentration of ~800 nM in the AIS (*Figure 9a,b*). Due to the high density of Na_V channels in the AIS they contribute to the majority of Δ[Ca²⁺]ᵢ and cause a rise of [Ca²⁺]ᵢ within submillisecond from the start of the AP (450 nM within <150 μs from AP threshold, red arrow in *Figure 9b*). These results are likely to provide an underestimate of the total Δ[Ca²⁺]ᵢ since in the model Ca²⁺ release from giant saccular organelle was not simulated, which would result in a total AP-evoked Δ[Ca²⁺]ᵢ of ~1.2 μM. In the basal dendritic branches the AP has a slower rise time and broader half-width, causing dendritic [Ca²⁺]ᵢ to accumulate slower and to higher concentrations, consistent with our experimental findings (*Figure 1*). Because the dendritic Na_V channel density is substantially lower, their contribution to the total [Ca²⁺]ᵢ is negligible. The distinct contribution of Na_V and Ca_V channels to [Ca²⁺]ᵢ is clearly visible when comparing the different Ca²⁺ currents in the AIS, showing that the majority of the total $I_{Ca}$ during an AP is carried by $I_{Ca(Na)}$ (*Figure 9c*). Simulations predict that the $I_{Ca(Na)}$ rapidly inactivates during the AP while $I_{Ca}$ activates more slowly and has an incomplete inactivation during the AP

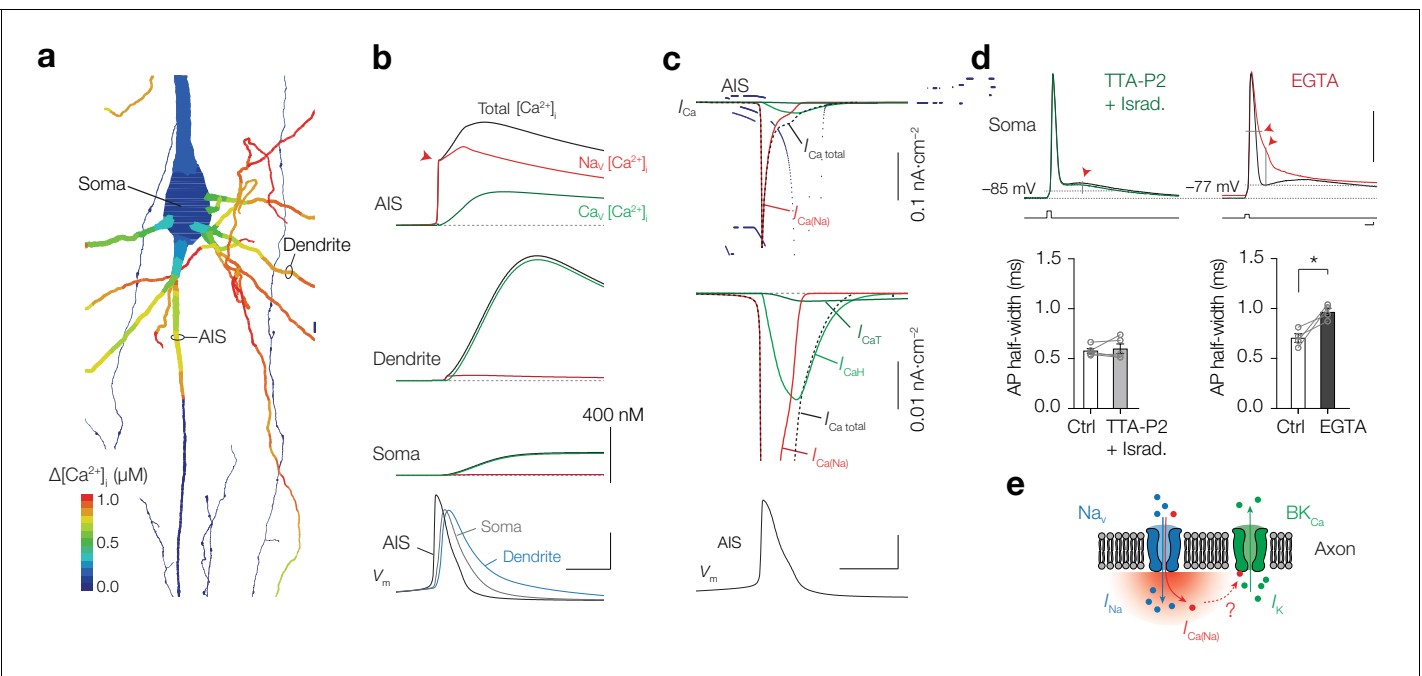

**Figure 9.** Na_V channels produce submillisecond near-micromolar [Ca²⁺]ᵢ at the AIS. (**a**) Color coded shape plot of simulated [Ca²⁺]ᵢ in the axon, dendrites and soma activated by an AP in current clamp without Ca²⁺ indicator. (**b**) The modeled [Ca²⁺]ᵢ in the AIS, Soma and Dendrite (locations specified in **a**). The total [Ca²⁺]ᵢ (black) is divided in a Na_V (red) and Ca_V (green) channel mediated fraction. *Bottom*, $V_m$ in the same compartments. Scale bars, 400 nM for all [Ca²⁺]ᵢ plots, 50 mV and 1 ms. (**c**) *Top*, $I_{Ca}$ in response to an AP (*bottom*) in the AIS. *Middle*, $I_{Ca}$ on expanded scale. Ca²⁺ extrusion contributes to $I_{Ca}$ total, but is not shown. Scale bars bottom, 50 mV and 1 ms. (**d**) *Top*, example traces of somatic APs in control (black) and after bath application of TTA-P2 and Isradipine (green, left) and EGTA (red, right). Resting membrane potential and AP voltage threshold are indicated by the dotted grey lines. Red arrows indicate the significant changes in ADP (with TTA-P2 + Isradipine) and AP width and AHP (with EGTA). Capacitance transients are removed for clarity. Scale bars indicate 50 mV, 2.5 nA, 1 ms. *Bottom*, population data of AP half-width before and after blocking T– and L–type Ca_V channels (*left*, p=0.61, two-tailed paired t-test, n = 5) and before and after preventing Ca²⁺ influx by bath application of EGTA (*right*, *p=0.035, two-tailed paired t-test, n = 4), see also *Table 1*. (**e**) Schematic of Ca²⁺ permeation through a Na_V channel activating a Ca²⁺–dependent BK channel.

**Table 1.** Effect of Ca²⁺ entry on AP waveform.

| | RMP (mV) | AP threshold (mV) | AP amplitude (mV) | AP half-width (µs) | AHP (mV) | ADP (mV) |
|---|---|---|---|---|---|---|
| Control | −77.4 ± 2.5 | −67.1 ± 3.0 | 110.9 ± 12.5 | 612.7 ± 24.7 | 5.6 ± 2.1 | 7.6 ± 2.5 |
| TTA-P2 + isradipine | −76.0 ± 2.9 | −65.0 ± 4.1 | 109.3 ± 3.3 | 634.5 ± 49.6 | 3.2 ± 2.9 | 4.6 ± 3.3 |
| Paired t-tests (n = 5) | p=0.32 | p=0.30 | p=0.33 | p=0.61 | p=0.055 | p=0.039* |
| Control | −70.4 ± 1.6 | −59.3 ± 3.9 | 102.4 ± 1.8 | 753.8 ± 45.9 | 3.7 ± 3.8 | 6.5 ± 3.4 |
| EGTA | −71.8 ± 3.6 | −65.4 ± 1.8 | 103.5 ± 2.4 | 1031.5 ± 39.6 | 25.5 ± 4.3 | 11.7 ± 1.5 |
| Paired t-tests (n = 4) | p=0.69 | p=0.19 | p=0.27 | p=0.035* | p=0.043* | p=0.17 |

Overview of mean and s.e.m. of AP properties compared between control and toxin experiments recorded at the soma. RMP, resting membrane potential, AHP, fast afterhyperpolarization, ADP, afterdepolarization. The APs were elicited by large and brief current injections (~6 nA for 0.5 ms) to obtain temporally aligned APs between trials and image OGB-1 fluorescence (**Figure 3**). AP amplitude, AHP and ADP were measured relative to the AP threshold. If the AHP or ADP was not detectable as a local peak, the membrane potential at the time point as in control was used (see EGTA example **Figure 9d**). P values are results of two-tailed paired t-tests before and after toxin application. Data available in **Table 1—source data 1**.

The online version of this article includes the following source data for Table 1:

Source data 1. Effect of Ca²⁺ entry on AP waveform.

repolarization, likely becoming the dominant contribution to $[Ca^{2+}]_i$ during the afterdepolarization and high-frequency spike generation (**Figure 9c** and **Figure 8—figure supplement 1**).

Our experiments and computational simulations show that $[Ca^{2+}]_i$ changes mediated by $Ca_V$ or $Na_V$ channels act at distinct spatiotemporal scales. To experimentally test the differential impact of $Ca^{2+}$ influx on the AP waveform, we analyzed the somatically recorded APs when using distinct blockers most effective in modulating $[Ca^{2+}]_i$ in the AIS (**Figure 3**). Blocking both T- and L-type $Ca_V$ channels, contributing to ~27% to AP-evoked $Ca^{2+}$ at the AIS (**Figure 3**), significantly reduced the afterdepolarization and showed a trend to reduce the afterhyperpolarization (ADP, p=0.039 and AHP, p=0.055, respectively, two-tailed paired t-tests, n = 5), without affecting other AP properties (p>0.30; **Figure 9d** and **Table 1**). In contrast, when lowering $[Ca^{2+}]_o$ with EGTA, which abolished all AIS $Ca^{2+}$ influx (**Figure 3**), the AP half-width significantly increased and the AHP was reduced (AP half-width, p=0.035 and AHP, p=0.043, two-tailed paired t-tests, n = 4; **Figure 9d**, **Table 1**). These results are consistent with the temporal differences in AP-evoked AIS $[Ca^{2+}]_i$ and suggest that $Na_V$-mediated $Ca^{2+}$ entry may act to open $BK_{Ca}$ channels (**Figure 9e**), thereby driving K⁺ efflux and facilitating axonal AP repolarization.

## Discussion

In the present study we identified $Ca^{2+}$ permeation through $Na_V$ channels as a source for activity-dependent $Ca^{2+}$ entry in mammalian axons. The findings were supported by independent and converging lines of evidence, ranging from anatomical compartmentalized $[Ca^{2+}]_i$ transients at sites with high $Na_V$ channel densities, a pharmacological block by TTX, an overlap of optically recorded $I_{Ca}$ with $Na_V$ channel gating and molecular evidence for $Ca^{2+}$ influx mediated by the $Na_V1.2$ channel. In axonal domains $Na_V$ channels are thus not only involved in electrically generating the upstroke of the action potential, but also contribute to cytoplasmic $Ca^{2+}$ signaling.

$Ca^{2+}$ entry in the L5 pyramidal neuron AIS was in part mediated by T- and L-type $Ca_V$ channels (**Figure 3** and **Figure 5**) in keeping with previous studies showing that $Ca_V$ channel subtypes mediate activity-dependent $Ca^{2+}$ changes in both central- and peripheral nervous system axons (**Callewaert et al., 1996**; **Bender and Trussell, 2009**; **Yu et al., 2010**; **Gründemann and Clark, 2015**; **Zhang and David, 2016**; **Clarkson et al., 2017**). However, when quantifying the specific fraction of block $Ca_V$ channels explained only ~35% of the total $Ca^{2+}$ entry during a single AP (**Figure 3**). These results are consistent with recent 2-photon $Ca^{2+}$ imaging from the prefrontal cortical pyramidal neuron AIS, showing that ~70% of the $[Ca^{2+}]_i$ transients evoked by a train of three APs remains in the presence of T-type $Ca_V$ channel block (**Clarkson et al., 2017**). Here, we found that for APs the remaining ~65% of $[Ca^{2+}]_i$ increase is actually TTX-sensitive and this accounted even for >90% of the

subthreshold-induced $[Ca^{2+}]_i$ changes (*Figure 4* and *Figure 5*). That $Ca^{2+}$ enters through $Na_V$ channels builds on landmark studies showing that the initial component of depolarization-induced $Ca^{2+}$ entry in the squid giant axon is tetrodotoxin (TTX)-sensitive (*Baker et al., 1971*; *Meves and Vogel, 1973*; *Brown et al., 1975*). Squid axons even generate rapid spikes in the sole presence of $Ca^{2+}$ ions (*Watanabe et al., 1967*). A TTX-sensitive $Ca^{2+}$ current has also been identified in hippocampal neurons (*Akaike and Takahashi, 1992*) and more extensively investigated in cardiac myocytes (*Aggarwal et al., 1997*; *Santana et al., 1998*; *Cruz et al., 1999*; *Heubach et al., 2000*; *Chen-Izu et al., 2001*). The incomplete selectivity of toxins and blockers for $Ca^{2+}$ channels continued, however, to cast doubt about the precise identity of the TTX-sensitive $Ca^{2+}$ current (*Cruz et al., 1999*; *Chen-Izu et al., 2001*; *Sun et al., 2008*). Indeed, an alternative explanation for some of the present results is that TTX blocks a $Ca_V$ channel subtype. This has been reported at very high concentrations (30 µM) for $Ca_V3.3$ (*Sun et al., 2008*), which is higher than what we used (1 µM). Although we cannot exclude the presence of a TTX-sensitive $Ca_V$ channel at the AIS that is not blocked by one of the compounds used in the pharmacological screening (*Figure 3*), our optically recorded $I_{Ca}$ provides biophysical evidence that the TTX-sensitive $Ca^{2+}$ current follows the same rapid activation time course as the $Na^+$ channel pore, incompatible with T-type channel kinetics (*Figure 6*). Importantly, in further support that $Na_V$ channels give rise to cytoplasmic $Ca^{2+}$ changes, heterologous expression of α- and β-subunits of $Na_V1.2$, known to be expressed in the rodent and human AIS (*Garrido et al., 2003*; *Hu et al., 2009*; *Tian et al., 2014*), showed that the channel proteins expressed in isolation were sufficient to mediate $Ca^{2+}$ influx (*Figure 7*). These results are in support of the findings by Lederer and colleagues showing that heterologous expression of $hNa_V1.5$ channel produces depolarization-evoked $[Ca^{2+}]_i$, if expressed with its β subunits (*Cruz et al., 1999*; *Guatimosim et al., 2001*).

From an evolutionary point of view, some $Ca^{2+}$ permeation of $Na_V$ channels is not surprising. $Na_V$ channels evolved from the $Ca_V$ channel superfamily and share molecular structure both in their pore sequence and intracellular regulatory domains (*Zakon, 2012*; *Ben-Johny et al., 2014*). Furthermore, the Born radii for $Na^+$ and $Ca^{2+}$ are comparable (1.68 and 1.73 Å, respectively) and $Ca^{2+}$ ions are known to enter the $Na_V$ channel pore to block $Na^+$ permeation in a concentration-dependent manner (*Lewis, 1979*; *Armstrong and Cota, 1999*). Interestingly, single residue mutations in the selectivity filter of $Na_V$ channels suffices to increase $Ca^{2+}$ ion permeation (*Heinemann et al., 1992*; *Naylor et al., 2016*). Our calculations indicate that the conductivity of the $Na_V$ channel for $Ca^{2+}$ is ~0.4% (*Figure 7*). If we assume there exists proportionality between permeability and conduction we can apply the equation of permeability ~ conductance/concentration × valency$^2$ (*Baker et al., 1971*; *Meves and Vogel, 1973*). With our extracellular solutions $[Ca^{2+}]_o/[Na^+]_o$ being 0.0148 and the valence ratio $(Ca^{2+}/Na^+)^2$ being four we can calculate that $Na_V$ channels in mammalian axons have a $P_{Ca}/P_{Na}$ ratio of 0.06. Notably, the value is in range of direct recordings for $P_{Ca}/P_{Na}$ in the sciatic nerve and squid axons (0.10 and 0.14, *Hille, 1972*; *Meves and Vogel, 1973*) as well as recordings from $Na_V1.5$ channels revealing $P_{Ca}/P_{Na}$ ratios of ~0.04 (*Cruz et al., 1999*). Such permeability is orders of magnitude lower than $P_{Ca}/P_{Na}$ ratios of acetylcholine receptors or NMDA receptors (1.0 and 17, respectively) (*Lewis, 1979*; *Iino et al., 1997*). An independence of $Na^+$ and $Ca^{2+}$ ions, modeled as ohmic conductances, will be a major simplification of the $Na_V$ channel under multi-ion conditions. Molecular dynamic studies of $Na_V$ channels showed that ionic interactions between $Na^+$ and $Ca^{2+}$ at the channel pore are complex (*Corry, 2013*; *Boiteux et al., 2014*; *Naylor et al., 2016*). The energy barrier of the selectivity filter strongly favors $Na^+$ ions but can be flexible, changing in conformational states and consistent with modest $Ca^{2+}$ permeation (*Corry, 2013*; *Boiteux et al., 2014*; *Naylor et al., 2016*). In future experiments it will be interesting to obtain more detailed permeability ratios ($P_{Ca}/P_{Na}$) by recording changes in $E_{rev}$ with varying intra- and extracellular concentrations, fitting the data to mathematical solutions such as the electro-diffusion theory of Goldman-Hodgkin-Katz (GHK) extended with surface charge potentials (*Lewis, 1979*; *Campbell et al., 1988*), or using Eyring–Läuger theory based on individual ionic rate constants (*Läuger, 1973*). Such experiments would in particular be interesting for $Na_V1.6$, the main isoform expressed in axonal domains (*Lorincz and Nusser, 2010*; *Kole and Stuart, 2012*).

Although the $Ca^{2+}$ conductivity of the channels is small it achieves near–micromolar $Ca^{2+}$ changes in axons as $Na_V$ channels are clustered to very high densities (~1000 channels $\mu m^{-2}$) at the AIS and nodes of Ranvier (*Neumcke and Stämpfli, 1982*; *Lorincz and Nusser, 2010*; *Kole and Stuart, 2012*). Consistent with this idea, our imaging experiments showed that also nodes of Ranvier

produced subthreshold-activated $Ca^{2+}$ entry (*Figure 1* and *Figure 4*), suggesting that $Na_V$ channel mediated $Ca^{2+}$ entry could play similar roles in these domains. At the AIS, the rapid opening of high densities of $Na_V$ channels may further act as a trigger to amplify $[Ca^{2+}]_i$ via the activation of ryanodine receptors, mediating ER store release of $Ca^{2+}$ from the giant saccular organelle, which extends continuously along the AIS of thick-tufted L5 pyramidal neurons (*Figure 2*).

The rapid inactivation of $Na_V$ channels compared to the slow inactivation of $Ca_V$ channels will lead to voltage- and time-dependent changes in the relative contribution of $Na_V$ and $Ca_V$ channels to $[Ca^{2+}]_i$. In axons, a single AP will mostly lead to $Na_V$-mediated $Ca^{2+}$ entry while an increasing number of APs, or longer sustained depolarization, will lead to an accumulation of $Ca^{2+}$ mediated by $Ca_V$ channel activation. Indeed, $Ca^{2+}$ entry via $Ca_V$ channels has been identified as a major contributor to $Ca^{2+}$ entry in sciatic nerve or Purkinje axons during trains of APs or prolonged depolarization for hundreds of milliseconds (*Callewaert et al., 1996*; *Zhang and David, 2016*). However, in vivo recordings from L5 pyramidal neurons show that they typically fire sparsely and on average ~1–4 Hz (*de Kock et al., 2007*) and the half-width of axonal APs is about ~300 µs (*Kole et al., 2007*). In this view, $Na_V$-mediated $Ca^{2+}$ entry may be the main source for activity-dependent $[Ca^{2+}]_i$ in the excitable domains of axons under physiological conditions.

What could be the functional role of $Na_V$-mediated $Ca^{2+}$ entry in axon initial segments and nodes of Ranvier? One downstream target of submembranous axoplasmic $[Ca^{2+}]_i$ may be regulation of $Na_V$ inactivation kinetics via their $Ca^{2+}$/calmodulin domain at their C-terminus as has been demonstrated for multiple $Na_V$ subtypes, including $Na_V1.2$ and $Na_V1.6$ (*Sarhan et al., 2012*; *Reddy Chichili et al., 2013*; *Ben-Johny et al., 2014*; *Wang et al., 2014*). Another target of $Na_V$-mediated $Ca^{2+}$ could be to open axonal large-conductance $BK_{Ca}$ channels. The $BK_{Ca}$ channel opens with the cooperative action of membrane voltage and $[Ca^{2+}]_i \geq 10$ µM to repolarize APs and shorten their duration (*Berkefeld et al., 2010*). Across cell types there are considerable variations in the magnitude and time course of $BK_{Ca}$ currents due to differences in nanodomain coupling with $Ca_V$ channel isoforms. $BK_{Ca}$ channels are exclusively activated by P/Q-type $Ca_V$ channels in cerebellar Purkinje neurons with short AP durations (*Womack et al., 2004*), but in rat chromaffin cells with wider APs, $BK_{Ca}$ channels are coupled with the Q- and slower activating L-type $Ca_V$ channels (*Prakriya and Lingle, 1999*). Also in L5 pyramidal neurons $BK_{Ca}$ activation shortens the duration of somatic APs from ~1 ms to ~600 µs (*Yu et al., 2010*; *Bock and Stuart, 2016*; *Roshchin et al., 2018*). Considering the brief duration of axonal APs in the L5 pyramidal neurons (~300 µs, *Kole et al., 2007*) $Na_V$ channels may provide both a precisely-timed voltage-dependent activation, via $Na^+$ current, as well as a $[Ca^{2+}]_i$ rise within 150 µs (*Figure 9*), to rapidly open $BK_{Ca}$ channels and shape axonal AP repolarization. In agreement with this conjecture our data show that T- and L-type $Ca_V$ channels are too slow to mediate somatic AP repolarization, leaving open the possibility that a $Na_V$-$BK_{Ca}$ channel nanodomain coupling provides the required $Ca^{2+}$ signal. Firm evidence for such interaction would require mutating the selectivity filter of $Na_V$ channels to abolish $Ca^{2+}$ but not $Na^+$ permeation. Recent two-photon $Ca^{2+}$ uncaging experiments already showed that the $[Ca^{2+}]_i$ rise at the first node of Ranvier in L5 axons opens nodal $BK_{Ca}$ channels to shorten the AP duration and facilitate the generation of high firing rates in the proximal axon (*Roshchin et al., 2018*). Further downstream from the initiation site nodal $BK_{Ca}$ channels in Purkinje axons play a role in augmenting the hyperpolarization following APs and facilitate recovery from $Na_V$ inactivation to prevent propagation failures (*Hirono et al., 2015*). The dual permeation of axonal $Na_V$ channels for $Na^+$ and $Ca^{2+}$ ions may thus serve a common function; mediating the rapid electrical upstroke of the AP and via $Ca^{2+}$ signaling activating $K^+$ efflux to recover from inactivation and accelerating $Na_V$ channel availability for the next AP, representing a fine-tuning specifically to the needs of axonal AP generation and conduction fidelity.

## Materials and methods

**Key resources table**

| Reagent type (species) or resource | Designation | Source or reference | Identifiers | Additional information |
|---|---|---|---|---|
| Gene (human) | SCN2A | Genscript, USA (*Ben-Shalom et al., 2017*) | | |
| Gene (human) | SCN1B/SCN2B | Genscript, USA (*Ben-Shalom et al., 2017*) | | |

*Continued on next page*

*Continued*

| Reagent type (species) or resource | Designation | Source or reference | Identifiers | Additional information |
|---|---|---|---|---|
| Cell line (Homo-sapiens) | HEK-293T/17 | ATCC | Cat#: CRL-11268, RRID:CVCL_1926 | |
| Commercial assay or kit | GeneJET Plasmid Maxiprep kit | ThermoFisher, USA | Cat#: K0491 | |
| Chemical compound, drug | Ryanodine | Tocris | Cat#: 1329 | |
| Chemical compound, drug | Heparin | Tocris | Cat#: 2812 | |
| Chemical compound, drug | TTA-P2 | Alomone | Cat#: T-155 | |
| Chemical compound, drug | Isradipine | Tocris | Cat#: 2004 | |
| Chemical compound, drug | Nickel | Sigma | Cat#: N6136 | |
| Chemical compound, drug | Nimodipine | Tocris | Cat#: 0600/100 | |
| Chemical compound, drug | SNX-482 | Tocris | Cat#: 2945 | |
| Chemical compound, drug | ω-conotoxin MVIIC | Tocris | Cat#: 1084 | |
| Chemical compound, drug | KB-R7943 | Tocris | Cat#: 1244 | |
| Chemical compound, drug | SN-6 | Tocris | Cat#: 2184 | |
| Chemical compound, drug | TTX | Tocris | Cat#: 1069 | |
| Chemical compound, drug | QX-314 | Alomone | Cat#: Q-150 | |
| Chemical compound, drug | EGTA | Sigma | Cat#: E0396 (low extracellular $Ca^{2+}$), Cat#: 3777 (HEK-cell experiments) | |
| Chemical compound, drug | CNQX | HelloBio | Cat#: HB0205 | |
| Chemical compound, drug | D-AP5 | HelloBio | Cat#: 0225 | |
| Chemical compound, drug | ZD-7288 | Tocris | Cat#: 1000 | |
| Chemical compound, drug | XE991 | Tocris | Cat#: 2000 | |
| Chemical compound, drug | Gabazine | Sigma | Cat#: S106 | |
| Chemical compound, drug | OGB-5N | Invitrogen | Cat#: O6812 | |
| Chemical compound, drug | OGB-1 | Invitrogen | Cat#: O6806 | |
| Chemical compound, drug | Bis-Fura2 | Biotium | Cat#: 50045 | Discontinued |
| Chemical compound, drug | SBFI | Invitrogen | Cat#: 10033152 | |
| Chemical compound, drug | JPW3028 | Potentiometric Probes | JPW3028 | |

*Continued on next page*

*Continued*

| Reagent type (species) or resource | Designation | Source or reference | Identifiers | Additional information |
|---|---|---|---|---|
| Antibody | anti-synaptopodin (Rabbit polyclonal) | Sigma | Cat#: S9442, RRID:AB261570 | (1:500) |
| Antibody | Anti-ankyrinG (mouse monoclonal) | Neuromab | Cat#: 75–146, RRID:AB_10673030) | (1:100) |
| Antibody | Anti-βIV spectrin (mouse monoclonal) | Neuromab | Cat#: 75–377, RRID:AB_2315818) | (1:250) |
| Other | streptavidin Alexa-488 conjugate | Invitrogen | Cat#: S32354, RRID:AB_2315383 | (1:500) |
| Software, algorithm | Neuroplex | RedShirt Imaging | RRID:SCR_016193 | |
| Software, algorithm | Axograph | Axograph | RRID:SCR_014284 | Version 1.7.0 |
| Software, algorithm | GraphPad Prism | GraphPad Prism | RRID:SCR_002798 | Version 8.4.2 |
| Software, algorithm | FIJI | *Schindelin et al., 2012* | RRID:SCR_002285 | |
| Software, algorithm | µManager | *Edelstein et al., 2014* | RRID:SCR_016865 | |
| Software, algorithm | NEURON | *Hines and Carnevale, 2001* | RRID:SCR_005393 | |
| Software, algorithm | Maxchelator | *Bers et al., 2010* | RRID:SCR_000459 | |
| Software, algorithm | FrameSplitter | *Battefeld et al., 2018* | https://github.com/Kolelab/Image-analysis | |

## Ethical approval

All animal experiments were performed in compliance with the European Communities Council Directive 2010/63/EU effective from 1 January 2013. They were evaluated and approved by the national CCD authority (license AVD8010020172426) and by the KNAW animal welfare and ethical guidelines and protocols (DEC NIN 14.49, DEC NIN 12.13, IvD NIN 17.21.01 and 17.21.03). Written informed consent was obtained from patients and all procedures on human tissue were performed with the approval of the Medical Ethical Committee of the Amsterdam UMC, location VuMC and in accordance with Dutch license procedures and the Declaration of Helsinki. All data were anonymized.

## Tissue collection

Young-adult male Wistar rats (RjHan:WI) were used at an age between P21 and P35 (Charles River Laboratories and Janvier labs). Animals were deeply anaesthetized by 3% isoflurane inhalation, decapitated and 300 µm parasagittal slices containing the primary somatosensory cortex were cut with a Vibratome (1200S, Leica Microsystems B.V.) within ice-cold artificial cerebrospinal fluid (ACSF) of the following composition (in mM): 125 NaCl, 3 KCl, 25 glucose, 25 NaHCO$_3$, 1.25 Na$_2$H$_2$PO$_4$, 1 CaCl$_2$, 6 MgCl$_2$, saturated with 95% O$_2$ and 5% CO$_2$ (pH 7.4). Following a recovery period at 35°C for 35–45 min slices were stored at room temperature in the ACSF. Human slices were obtained from non-pathological cortex removed for the surgical treatment of deeper brain structures for mesial temporal lobe epilepsy. After resection, a block of the temporal lobe was placed within 30 s in ice-cold artificial cerebrospinal fluid (ACSF) slicing solution which contained in (mM): 110 choline chloride, 26 NaHCO$_3$, 10 D-glucose, 11.6 sodium ascorbate, 7 MgCl$_2$, 3.1 sodium pyruvate, 2.5 KCl, 1.25 NaH$_2$PO$_4$, and 0.5 CaCl$_2$ (300 mOsm) and transported to the laboratory, as described in detail previously (*Testa-Silva et al., 2014*). Transition time between resection of the tissue and preparation of the slices was <15 min. Neocortical slices (~350 µm thickness) were cut in an ice-cold slicing solution, stored for 30 min at 34°C, and afterwards switched to room temperature in standard ACSF. Slices were subsequently transported (<15 min) towards the NIN (KNAW) in continuously carbogenated ACSF.

## Cell lines

Human embryonic kidney 293 cells (HEK 293T/17 cell line, CRL-11268 obtained from ATCC) were cultured in growth medium consisting of equal parts of Dulbecco's modified Eagle's medium (DMEM) (DMEM Glutamax, Gibco, Thermo Fisher Scientific) and Ham's F10 nutrient mix (Gibco,

Thermo Fisher Scientific), supplemented with 10% fetal calf serum (FCS) and 1% penicillin–strepto-mycin. Cells were split twice a week by trypsinization and grown at 37°C with a humidified atmosphere containing 5% $CO_2$. STR profiling confirmed a 100% match with the HEK 293T cellline (ATCC). Human *SCN2A* (D-splice variant), encoding for the alpha subunit of the Na$_V$1.2 channel was cloned in pcDNA3.1-IRES-GFP, and *SCN1B/SCN2B*, encoding for beta subunits 1 and 2, was cloned into pcDNA3.1. These vectors were described previously (*Ben-Shalom et al., 2017*) and obtained from Genscript (Genscript, USA). The constructs were amplified in Stbl3 bacteria (Genscript, USA) and were purified using the GeneJET Plasmid Maxiprep kit (ThermoFisher, USA) according to the manufacturer's protocols. The plasmids were transiently transfected into 70% confluent HEK-293 cells plated in 12-well plates. Per well, the transfection cocktail contained 500 ng pcDNA3.1-SCN2A-IRES-GFP, 290 ng pcDNA3.1-SCN1B- IRES-SCN2B and 5 µL of polyethylenimine (PEI) diluted in 100 µL 1% saline, incubated for 20 min at room temperature before addition to the culture medium. Cells were incubated with 100 µL of transfection cocktail in 1 mL of culture medium for 24 hr at 37°C in a humidified atmosphere containing 5% $CO_2$. Cells were trypsinised and used for electrophysiological recording typically 48 hr after transfection.

## Electrophysiological recording from neurons

For patch-clamp recording, slices were transferred to a customized upright microscope (BX51WI, Olympus Nederland BV, or LNscope, Luigs and Neumann, Ratingen, Germany). The transmitted light path consisted of a custom made 850 nm Light Emitting Diode (LED) light source (LZ1-10R602, LED Engin, CA), collimated using an aspheric condenser lens (ACL50832U-B, Thorlabs, Germany) and adapted to the microscope lamp port using a custom 3D printed adapter and passed through an oblique or Dodt illumination condenser (WI-OBCD, Olympus). The top 50 µm of the slice surface was visualized using an optical pathway consisting of a 60× water immersion objective (N.A. 1.0, LUMPLFLN60XW, Olympus or N.A. 1.1, LUMPLFLN60XW, Olympus), 2× intermediate zoom attachment (U-ECA, Olympus), camera splitter (U-TRU, Olympus) with inbuilt 180 mm tube lens on the back port and a 0.63× demagnifier (U-TV0.63XC, Olympus) projected the final image onto a high resolution CCD camera (CoolSNAP-EZ, Photometrics), which was operated using µManager (*Edelstein et al., 2014*). Based on the bright-field image large L5 neurons with an intact axon parallel and close to the surface were targeted for recording. Current-clamp recordings were made with Dagan BVC-700A amplifiers (Dagan Corporation, MN, USA) or AxoClamp 900A (Molecular Devices Limited, UK). An Axopatch 200B (Molecular Devices) was used for voltage-clamp and AP-clamp experiments. The microscope bath was perfused with oxygenated (95% $O_2$, 5% $CO_2$) ACSF consisting of (in mM): 125 NaCl, 3 KCl, 25 glucose, 25 NaHCO$_3$, 1.25 Na$_2$H$_2$PO$_4$, 2 CaCl$_2$, and 1 MgCl$_2$. Patch pipettes were pulled from borosilicate glass (Harvard Apparatus, Edenbridge, Kent, UK) pulled to an open tip of 3–6 MΩ resistance. For all current-clamp, subthreshold voltage-clamp ramp and AP-clamp recordings the intracellular solution contained (in mM): 130 K-Gluconate, 10 KCl, 4 Mg-ATP, 0.3 Na$_2$-GTP, 10 HEPES and 10 Na$_2$-phosphocreatine (pH 7.25 adjusted with KOH, 280 mOsmol kg$^{-1}$). The liquid junction potential difference of –13.5 mV was corrected in all recordings. For morphological reconstruction, 5 mg ml$^{-1}$ biocytin was routinely added. Voltage recordings were analogue low-pass filtered at 10 kHz (Bessel) and digitally sampled at 100 kHz using A-D converter (ITC-18, HEKA Elektronik Dr. Schulze GmbH, Germany) and data acquisition software Axograph X (v.1.5.4, Axograph Scientific, NSW, Australia). Bridge-balance and capacitances were fully compensated based on small current injections leading to minimal voltage errors. The recording temperature was 33 ± 1°C. Only cells with a stable bridge-balance (<25 MΩ), resting membrane potential and AP shape throughout the recording session were included in the analysis.

For voltage-clamp recordings of $I_{Na}$ and $I_{Ca}$ (*Figure 5*) the bath was perfused with oxygenated (95% $O_2$, 5% $CO_2$) extracellular recording solution consisting of (in mM): 100 NaCl, 3 KCl, 25 glucose, 25 NaHCO$_3$, 1.25 Na$_2$H$_2$PO$_4$, 2 CaCl$_2$, 1 MgCl$_2$, 5 4-AP, 20 TEA-Cl, 0.02 CNQX, 0.05 D-AP5, 0.02 ZD-7288, 0.01 XE991 and 0.003 Gabazine (SR-95531). The intracellular solution contained (in mM): 130 CsCl, 10 TEA-Cl, 10 HEPES, 4 Mg-ATP, 5 Na$_2$-phosphocreatine and 0.3 Na$_2$-GTP (pH 7.25 adjusted with CsOH, 280 mOsmol kg$^{-1}$). A liquid junction potential of –5.6 mV was applied to the recordings. Series resistance was routinely compensated to >75% and the linear leak and capacitance off-line subtracted using a P/9 protocol with 10-fold scaled pulses. Current recordings were analogue low-pass filtered at 10 kHz (Bessel) and digitally sampled at 100 kHz using A-D converter (ITC-18, HEKA Elektronik Dr. Schulze GmbH, Germany) and data acquisition software Axograph X

(v.1.5.4, Axograph Scientific, NSW, Australia). To improve voltage-clamp of the large and rapid Na$^+$ currents the recordings were made at room temperature (~20°C).

## Electrophysiological recordings from HEK-293 cells

For recordings from HEK-293 cells they were transferred to a recording chamber which was continuously perfused with extracellular solution, containing (in mM): 135 NaCl, 4.5 KCl, 2 CaCl$_2$, 1 MgCl, 10 HEPES and 11 Glucose. The intracellular solution contained (in mM): 110 CsF, 10 NaCl, 20 EGTA and 10 HEPES. In the OGB-1 experiments, we added 100 µM OGB-1, EGTA was omitted and CsF raised to 120 mM instead of 110 mM. The liquid junction potential difference of –10 mV was corrected for. Whole-cell patch-clamp recordings were made ~48 hr after transfection. Cells were recorded at room temperature (~20°C) and continuously perfused with extracellular solution at a flow rate of 1.5 mL•min$^{-1}$. Patch pipettes were pulled to a resistance of 2–3 MΩ. Round, isolated cells with a diameter >10 µm, a smooth cell surface and a moderate EGFP fluorescent signal were selected for recordings (*Figure 7a*). HEK-293 cells had an average capacitance of 9.19 ± 0.70 pF (*n* = 16). The holding potential was –70 mV and voltage dependence of activation of Na$_V$1.2 was determined by an activating protocol consisting of a hyperpolarizing pulse to –130 mV (20 ms) followed by step pulses from –80 mV to +50 mV with increments of 10 mV for 20 ms. Voltage dependence of inactivation was assessed with voltage pulses from –130 mV to –30 mV with increments of 10 mV for 100 ms duration, followed by a depolarizing pulse to –20 mV for 20 ms. A P/5 leak subtraction protocol (10-fold scaling) was used to subtract remaining capacitive and leak currents. Series resistance was not compensated.

## Blockers and toxins

EGTA and blockers were added to the appropriate concentration to the ACSF and perfused. The extracellular Ca$^{2+}$ ([Ca$^{2+}$]$_o$) was lowered by bath application of 2.5 mM EGTA and using the online maxchelator tool (https://somapp.ucdmc.ucdavis.edu/pharmacology/bers/maxchelator/CaMgAT-PEGTA-TS.htm; *Bers et al., 2010*) we calculated the remaining [Ca$^{2+}$]$_o$ to be 437 nM, based on a recording temperature of 35°C, a pH of 7.4 and an ionic strength of 0.15 M of the free ions in our solution. To limit hyperexcitability in the presence of EGTA, we added synaptic blockers to the ACSF (20 µM CNXQ and 50 µM D-AP5) and kept the a-EPSP voltage peak amplitude constant by reducing the amplitude of the current injections in both control and EGTA measurements (*Figure 2*). To prevent precipitation of Ni$^{2+}$ we used phosphate-free extracellular solutions containing (in mM): 126.25 NaCl, 3 KCl, 25 glucose, 25 NaHCO$_3$, 2 CaCl$_2$, 1 MgCl$_2$ and 0.1 Ni$^{2+}$. A > 10% increase in fluorescence baseline was observed in some experiments (3 out of 8 recordings), which were subsequently excluded. Two blockers (SNX-478 and ω-conotoxin MVIIC) were not perfused but were locally puffed using a Picospritzer III (Intracel) for 3 s ending 0.5 s before imaging to avoid vibration artifacts. Bovine serum albumin (0.1 mg/ml) was added to the rACSF before ω-conotoxin MVIIC was introduced to minimize non-specific binding of the drug.

## Ca$^{2+}$ and Na$^+$ imaging

To optically record [Ca$^{2+}$]$_i$ and [Na$^+$]$_i$ in axons, membrane impermeable Ca$^{2+}$ and/or Na$^+$ indicators were added to intracellular solutions. For Ca$^{2+}$ we used OGB-1 (100 µM), OGB-5N (1 mM) or bis-Fura-2 (200 µM) and for Na$^+$ imaging we used sodium-binding benzofuran isophthalate (SBFI, 1–1.5 mM). Patch pipettes were first filled with dye-free solution for half of the tapered part of the pipette tip, then backfilled with the dye-containing solution. Fluorescence intensity at the AIS was monitored during dye loading and imaging started only when the indicators were fully equilibrated (typically after 0.5–1 hr). Optical recordings of Ca$^{2+}$ or Na$^+$ dye fluorescence changes were obtained with wide-field epifluorescence microscopy. Fluorescence was collected by the same 60× water immersion objective, passed through the microscope tube lens (U-TR30IR, Olympus) and projected onto a rapid data-acquisition camera with relatively low spatial resolution (80 × 80 pixels) but high dynamic range (14 bits) and low read noise (NeuroCCD-SM, RedShirtImaging LLC, Decatur, GA) via a 0.1× or 0.35× demagnifier. The CCD frame corresponded to an area of approximately 320 or 91 µm$^2$ in the object plane with each individual pixel receiving light from an area of ~4 × 4 or 1.1 × 1.1 µm$^2$, respectively. High-speed recordings (20 kHz) were always performed with the 0.35× demagnifier and with 3 × 3 binning of pixels on the chip, the ultimate pixel sizes in these recordings were

~3.4 × 3.4 μm$^2$. The two recordings from human cells were performed under a 100× NA 1.1 Nikon objective (MRL07920) in combination with a 0.1× demagnifier, resulting in a pixel size of ~2.4 × 2.4 μm$^2$ (see *Figure 1—figure supplement 1*).

The epifluorescence light path consisted of an excitation LED light source, which was collimated using an aspheric lens (ACL5040U-A, Thorlabs, Germany) and the appropriate excitation filter, dichroic mirror and emission filter. For OGB-1 and OGB-5N excitation a 470 nm LED was used (SP-01-B4, Luxeon Star LEDs, Canada), the excitation light was filtered with 475/30 nm (475 nm center wavelength, 30 nm wide) excitation filter, reflected to the preparation by a dichroic mirror with a central wavelength of 500 nm and the fluorescence light was passed through a 520 nm barrier filter (U-MWB2 cube, Olympus). For SBFI excitation, LED light (365 nm LED, LZ1-10UV00, Ledengin, USA) was filtered by 357/44 nm filter (FF01-357/44-25, Semrock), a dichroic mirror with edge at 415 nm (Di03-R405-t1−25 × 36, Semrock) reflected excitation light to the sample and the emission light was then passed through a long-pass colored glass filter with the edge at 400 nm (FGL400, Thorlabs). For combined Na$^+$ and Ca$^{2+}$ imaging (*Figure 6—figure supplement 1*), the light from the 365 and 470 LEDs was combined by a dichroic mirror with edge at 458 nm (FF458-Di02, Semrock) and the filter set switched between trials. Light was directed through a fluorescence illuminator (BX-RFA, Olympus) equipped with a rectangular field stop providing an open area of 150 × 250 μm to reduce phototoxicity (U-RFSS, Olympus). The cell body was positioned just outside the field stop and the axon in the middle parallel to the long side (see e.g. *Figure 1*).

The critical benefit of epifluorescence measurements over two-photon imaging is increased light collection (~90% quantum efficiency, low-read noise of the CCD camera) enabling a high sensitivity and temporal fidelity. We optimized all imaging parameters to obtain maximal signal to noise ratio, which allowed us to image at the maximum acquisition rate of 20 kHz. In addition to the light collection optimization and selective targeting of superficial neurons, multiple trials were averaged to improve signal-to-noise ratio (typically 20–40). Fluorescence signals were temporally aligned to the electrophysiological voltage or current signals. For optical recording of $I_{Ca}$, which requires the transformation of ΔF/F into the first time derivative, Ca$^{2+}$ binding to the indicator must be proportional to [Ca$^{2+}$]$_i$ and the endogenous buffering capacity to be low. Based on the submillisecond equilibration time of OGB-5N and imaging at the maximally possible frame rate of 20 kHz it was recently shown that in CA1 hippocampal neurons these conditions are met and optically recorded $I_{Ca}$ tracks electrically recorded $I_{Ca}$, enabling the identification of Ca$_V$ channel subtypes in dendrites (*Jaafari et al., 2014*). Considering the low buffering capacities of endogenous buffers in the axon ($\kappa_s \approx 20$) (*Jackson and Redman, 2003*; *Delvendahl et al., 2015*), we employed this technique in the AIS. Ca$^{2+}$ imaging in HEK-293 cells was performed with 0.1 mM OGB1 added to the HEK-293 cell intracellular solution (from which EGTA was omitted). The fluorescence was recorded during a 200 Hz 1 s train of depolarizing pulses from –120 to –30 or –20 mV (corrected for liquid junction potential).

## Voltage imaging

Voltage imaging in neurons was performed as reported previously (*Hamada et al., 2017*). Neurons were filled with intracellular solution containing JPW3028 (0.8 mM) for typically 1 hr at room temperature, after which the patch pipette was retracted and the dye was left to diffuse into the lipid membranes for 1–4 hr. Subsequently the bath temperature was increased to 35°C and the cell was re-patched with normal intracellular solution. A 530 nm LED (SP-05-G4, Luxeon Star LEDs, Canada) was used for excitation of the dye. The excitation light was filtered with a 530/20 nm filter (BP510-550, Olympus), reflected to the sample by a dicroic mirror with a center wavelength of 570 nm (DM570, Olympus) and the emission light filtered by a 590 longpass filter (BA590, Olumpys). Data were collected at 20 kHz and low-pass filtered by a binomial filter (one pass) and averaged over 20–30 trials. Voltage imaging in HEK-293 cells was performed identically, with the only exception being that the experiments were performed at 20°C and the dye diffused equally in the small round cells, so imaging experiments were initiated 20 min after obtaining whole-cell configuration. The cells were maintained at –75 mV holding potential and the fluorescence of JPW3028 recorded at 1 kHz. The voltage command consisted of 100 ms steps of 50 mV increasing steps with a maximal step of +250 mV relative to holding potential. The average ΔF/F per voltage step was defined as the first 20 frames of each bleach corrected and normalized voltage step.

## Imaging data analysis

Imaging data analysis was performed using Neuroplex (Redshirt imaging), Axograph and Excel. Fluorescence signals were always background-subtracted. To correct for bleach effects, every 5th trial was recorded without current injection. A first order exponential was fitted to the average of the bleach trials and normalized to the peak. The average of the signal trials was divided by this trace to correct for bleach decay. Values for each ROI were defined as a fractional fluorescence change ($\Delta F/F_{baseline}$), where $F_{baseline}$ is the raw intensity average of 10 frames before the signal (subthreshold or AP) was initiated. Pixels were color coded with 'physics' color scheme from FIJI image processing software (NIH, USA) (*Schindelin et al., 2012*). For both OGB-1 and OGB-5N, we recorded the fluorescence in response to subthreshold stimuli, single APs and multiple APs. The $\Delta F/F$ response to subthreshold stimuli and a single AP was always below dye saturation. For OGB-1, we recorded trains of APs and observed that a single AP was 25 ± 2.7% of dye saturation (*n* = 8) and for OGN-5N, we observed a linear increase from 1 to 3 APs (*n* = 3), indicating that the fluorescence of a single AP was far from dye saturation.

## Optical current measurements

To compare kinetics between electrically and optically recorded currents, electrical currents were first downsampled to 20 kHz (optical acquisition rate). The optical trace was differentiated and then inverted (to mimic $I_{Na}$, which is conventionally depicted as a negative, inward current). All current traces were then filtered with a 3-window binomial filter of 50–150 iterations (generally 100). The traces were baselined to the current before the onset of the fast current and normalized to the peak of the current. Because the $I_{Ca\_opt}$ traces were too noisy to be fitted with a single exponential fit, conventionally used to obtain activation rise time, we used a Boltzmann sigmoid function to obtain the slopes of the traces.

$$y = bottom + \frac{(top - bottom)}{1 + e^{\left(\frac{x - x_0}{k}\right)}}$$

We fitted all traces using Axograph and the slope values (*k*) were used to compare activation kinetics. Although the $I_{Na}$ peak amplitude recorded at the soma highly varied between neurons (–36.61 ± 6.53 nA), the slope was nearly constant (207 ± 0.007 μs, *n* = 17).

## Ratiometric imaging

To estimate the absolute $[Na^+]_i$ and $[Ca^{2+}]_i$ in response to a single AP, we used a ratiometric imaging approach. Patch pipettes were front filled with clear intracellular solution and back-filled with intracellular solution containing 1.5 mM SBFI (Invitrogen) or 200 μM bis-Fura-2 hexapotassium salt (bF2, Biotium). Fluorescent emission of ratiometric indicators depends on the ionic concentration and the excitation wavelength where an increase in $[Ca^{2+}]_i$ produces an increase in bF2 fluorescence with the wavelength of 340 nm but a decrease with 385 nm. On the other hand, with SBFI an increase in $[Na^+]_i$ decreases SBFI fluorescence at 340 nm but does not alter at 385 nm excitation wavelength. Using the ratio (*R*) corrects for differences in cytosolic volume or dye concentration differences along imaged compartments (*Langer and Rose, 2009*). The sources of excitation light were two LEDs (Thorlabs) with peaks at 340 nm and 385 nm, fitted with band pass excitation filters at 340/22 and 387/11 nm (FF01-340/22-25 and FF01-387/11-25, Semrock) and combined by a dichroic mirror with a central wavelength of 376 nm (FF376-Di01−25 × 36, Semrock). The excitation light was reflected to the sample by a dichroic mirror with a central wavelength at 405 nm (Di01-R405−25 × 36, Semrock) and passed through the objective to the sample. The fluorescent emission signals were passed through a 420 long pass filter (Thorlabs). Ratiometric imaging was performed by alternatingly triggering each LED at the frame rate of the camera, as described previously (*Miyazaki and Ross, 2015*). This was achieved by combining custom designed Arduino/Parallax machines with Cyclops LED drivers. These hardware solutions allowed us to digitally control the voltage driving the LED, thus having maximum control over excitation light intensity. Fluorescence emission signals originating from each LED were separated with custom written software (FrameSplitter.txt, *Battefeld et al., 2018*). The camera operated at 0.5–1.0 kHz, resulting in a ratiometric frame rate of 0.25–0.5 kHz. Per experiment the fluorescent signals were averaged for 40 to 120 trials. The ratio R was defined as

$F_1/F_2$, where $F_1$ and $F_2$ are the background-subtracted fluorescence intensities at excitation with 340 nm and 385 nm, respectively.

## Calibration of ratiometric imaging

In order to scale ratiometric bF2 signals to absolute changes in $Ca^{2+}$ concentration, we used the standard equation for ratiometric measurements (equation 1 in *Figure 8—figure supplement 2*; *Grynkiewicz et al., 1985*), which depends on $K_D$, the dissociation constant, $R_{min}$ and $R_{max}$, the ratio in zero and dye-saturating $Ca^{2+}$, respectively and the scaling factor ($S_{f2}/S_{b2}$), defined as the fluorescence intensity at excitation with 385 nm of zero $Ca^{2+}$ divided by saturating $Ca^{2+}$. These values were determined in an ex situ calibration, by measuring the ratiometric signal of solutions containing 0 $Ca^{2+}$ and a high $[Ca^{2+}]$ (*Figure 8—figure supplement 2*). The solutions closely mimicked intracellular solutions and contained (in mM): 110 K-gluconate, 4.4 or 21 KCl, 0 or 10 $CaCl_2$, 3.8 or 5.36 $MgCl_2$, 10 HEPES, 4 Mg-ATP, 0.3 $Na_2$-GTP, 10 $Na_2$-phosphocreatine, 10 EGTA, and 0.2 bF2. The final free $[Ca^{2+}]$ depends on interaction between $Ca^{2+}$, $Mg^{2+}$ and EGTA and was calculated using the maxchelator tool (*Bers et al., 2010*).

$[Ca^{2+}]_{min}$ was 0 and $[Ca^{2+}]_{max}$ was 4.39 µM. We repeated the calibration experiment three times. In our experimental setting, the $K_D$ of bF2 was 507.3 ± 5.7 nM, which matches with the reported value of 525 nM in the presence of $Mg^{2+}$ (Molecular Probes Handbook, Thermofischer). We then performed ratiometric imaging in the AIS in response to a single AP. ΔR/R was calculated by dividing every ratio by the average of the baseline ratio before the onset of the AP (similar to the conventional ΔF/F). We used the $K_D$ as determined from our calibration experiments. The $R_{min}$ was scaled to be ~95% of $R_{baseline}$ to result in resting $[Ca^{2+}]_i$ of 50 nM and $R_{max}$ as established in the calibration experiments. If the LED intensity of the cellular recording was different from the calibration intensity used during the calibration experiments, the $R_{max}$ was corrected linearly, assuming that $R_{min}/R_{max}$ was constant. These experiments showed that after a single AP, $[Ca^{2+}]_{free}$ in the AIS rises with 55.6 ± 12.6 nM (see *Figure 8—figure supplement 2*).

Because $R_{min}$ and $R_{max}$ were not measured in situ we verified the Δ$[Ca^{2+}]_{free}$ with an alternative analysis that is independent of the exact values for $K_D$, $R_{min}$ and $R_{max}$ (equation 2 in *Figure 8—figure supplement 2*; *Langer and Rose, 2009*). In this approach, changes in fluorescence ratio ΔR/$R_0$ (%) are plotted versus $[Ca^{2+}]_{free}$, showing a nearly linear increase in ΔR/$R_0$ (%) for low $[Ca^{2+}]_{free}$ (see *Figure 8—figure supplement 2*). A linear fit to the region of $[Ca^{2+}]_i$ between 0 and 193 nM indicated that a 1% increase in ΔR/$R_{bF2}$ corresponded to a Δ$[Ca^{2+}]_i$ of ~10.4 nM ($R^2$ = 0.99, six concentrations, $n$ = 3 repetitions). We measured an AP-evoked Δ$[Ca^{2+}]_{free}$ of 52.5 ± 12.2 nM (*Figure 8—figure supplement 2*), in good support of the standard $Ca^{2+}$ measurement approach. We analyzed the ratiometric SBFI data only using this second approach, which is standard for SBFI measurements (*Langer and Rose, 2009*). The two base calibration solutions contained (in mM): 130 K-Gluconate or Na-Gluconate, 10 KCl of NaCl, 0.3 $Tris_2$-GTP or $Na_2$-GTP, 10 HEPES, 4 $Mg^{2+}$ATP, 10 $Tris_2$-phosphocreatine or $Na_2$-phosphocreatine, and 1.5 SBFI, pH 7.25 adjusted with Tris base. These two base solutions provided a range of 0–160.6 mM $[Na^+]$. When normalized to the ratio obtained in $Na^+$-free solution ($R_0$), a 1% increase in ΔR/$R_{SBFI}$ corresponded to a Δ$[Na^+]_i$ of 0.35 mM for Δ$[Na^+]_i$ between 0 and 48 mM ($R^2$ = 0.98; eight concentrations, $n$ = 3 repetitions, *Figure 8—figure supplement 2*). An AP evoked a Δ$[Na^+]_i$ of 1.49 ± 0.2 mM (*Figure 8—figure supplement 2*).

## Immunofluorescence staining

Following imaging experiments, the slices were fixed using 4% PFA in 0.1 M phosphate-buffered saline (PBS), pH 7.4 for 20 min and stored in 0.1 M PBS, pH 7.4 at 4℃. For triple immunohistological labeling the slices were washed three times in PBS and then incubated in a blocking solution (10% normal goat serum, 0.5% Triton X-100 in PBS) at room temperature for two hours, followed by 24 hr incubation at room temperature in the blocking solution containing primary antibodies: streptavidin Alexa-488 conjugate (1:500; Invitrogen), primary antibody for giant saccular organelle Synaptopodin (rabbit; 1:500; Sigma-Aldrich Chemie) and antibody for AIS marker: Ankyrin G (mouse; 1:100; Neuromab) or ßIV-spectrin (mouse; 1:250; Neuromab, see also Key Resources Table). The slices were 3x washed in 0.1 M PBS and then incubated with secondary antibodies: Alexa-555 goat anti rabbit IgG (1:500; Invitrogen) and Alexa 633 goat anti mouse IgG (1:500; Invitrogen). Subsequently, the slices were 3x washed in 0.1 M PBS and mounted with Vectashield mounting medium with 4,6-diamidino-

2-phenylindole (DAPI; Vector Laboratories). Images (bit depth, 8) were collected as described previously (*Hamada et al., 2016*). To align confocal images and the Ca$^{2+}$ fluorescence images of the Red-Shirt CCD camera (*Figures 1*, *2* and *4*) we used the original calibrated images of the two systems. The maximum Ca$^{2+}$ fluorescence image was calibrated within the original optical path. We overlaid the maximum Ca$^{2+}$ fluorescence image of the RedShirt camera and the maximum projection of the streptavidin image of the neuron morphology from confocal microscopy within ImageJ and applied only a rotation translation to visually match the two images based on the AP-evoked Ca$^{2+}$ signals spreading into dendrites and axons.

## Model simulations with single compartment

All model simulations were performed with NEURON (v.7.5) (*Hines and Carnevale, 2001*). A single compartment was created with length and diameter dimensions of 10 µm and *nseg* = 10, with specific membrane capacitance of 1.0 µF cm$^{-2}$, specific membrane resistance of 25 kΩ cm$^2$ and specific axial resistivity of 150 Ω cm. The resting membrane potential set to –77 mV using e_pass. Conductance models for Ca$^{2+}$ were based on the high-voltage activated (CaH) and a T-type Ca$_V$ channel model (CaT) obtained from ModelDB (https://senselab.med.yale.edu/ModelDB/) (*Mainen and Sejnowski, 1996*). Ca$^{2+}$ conductivity of Na$_V$ channels was modeled by including a standard ohmic Ca$^{2+}$ ion mechanism with a reversal potential (*eca*) of +140 mV into a mathematical 8-state Na$^+$ conductance model, computing simultaneously voltage- and time-dependence of the Ca$^{2+}$ current $I_{Ca(Na)}$ and $I_{Na}$ based on experimentally constrained rate constants of somatodendritic and axonal $I_{Na}$ (*Schmidt-Hieber and Bischofberger, 2010*). The kinetics of the voltage-gated conductance models was examined by fitting the current rise times with an exponential function for a –35 mV command potential, resampling the simulated traces to 20 kHz. The results showed that the $I_{Na}$ in the model activated with 240 µs and $I_{Ca(Na)}$ (0.5% conductivity ratio) with 280 µs. In comparison, $I_{CaT}$ activated with 4.88 ms and $I_{CaH}$ with 6.51 ms. These time constants are well in range of the experimentally determined values for the TTX- and Ni$^{2+}$-sensitive components recorded at the soma (*Figure 5*). For *Figure 8—figure supplement 1* we used an AP recorded from the L5 pyramidal neuron AIS at 100 kHz (threshold-to-peak, 94 mV; half-width duration of 285 µs [*Hallermann et al., 2012*] as the command potential in VectorPlay linked to the SEClamp function in NEURON (with $R_s$ being infinitely small). Single compartmental models were run at *dt* of 10 µs at a nominal temperature of 33°C.

## Model simulations with a multicompartmental model

Conductance-based multi-compartmental simulations were performed with an anatomically realistic reconstructed rat L5 pyramidal neuron (NeuroMorpho.Org ID: NMO_75667, Neuron Name 2014-04-01_1). The morphology was acquired with a confocal microscope at 2048 × 2048 pixels (1.0 µm *z*-steps, Leica SP8) using a 40× oil immersion objective (NA 1.3) scanning both the biocytin-streptavidin fluorescence and the ßIV-spectrin fluorescence. Uncompressed image stacks (~20 GB) were imported and reconstructed into Neurolucida (v.10, MBF Bioscience Inc, Germany), compartmentalized for the AIS and nodes as described previously (*Hamada et al., 2016*) and imported with the 3D import tool in NEURON (*Carnevale and Hines, 2006*). Multicompartmental simulations were performed to estimate the detailed ionic accumulation, concentration and diffusion in the proximal sites of the axon and match our experimental recordings as close as possible. Ca$^{2+}$ diffusion, buffering and pump (cdp) mechanisms were implemented based on the algorithms described in the NEURON book (Chapter 9, example 9.8 in *Carnevale and Hines, 2006*) and on a previously published Ca$^{2+}$ model (*Fink et al., 2000*) (available at ModelDB, accession number 125745, https://senselab.med.yale.edu/ModelDB/). We implemented cdp.mod (*Fink et al., 2000*) with the following alterations: we removed all SERCA related parameters, updated some starting values to our experimental conditions and extended the models to report not only [Ca$^{2+}$]$_i$, but also to simulate the Ca$^{2+}$ indicator response ΔF/F, using the equation:

$$\mathrm{F} = \frac{[dye]_{free} + c*[\mathrm{Ca}^{2+} + \mathrm{buffer}]}{[dye]_{total}}$$

With *c* being a constant to scale simulated ΔF/F. Because the equation was used to match the simulation to experimental data with regard to the temporal dynamics of Ca$^{2+}$ extrusion, the absolute amplitude of ΔF/F was not used and *c* was set to a nominal value of 6. The different Ca$^{2+}$

indicators used experimentally were implemented by adjusting the concentration of the exogenous buffer, and its known or measured $K_D$. Static $Ca^{2+}$ buffering properties of endogenous organelles ($\kappa_s$) were simulated with a TBufs of 100–400 µM and KDs of 10 µM, to mimic a $\kappa_s$ of 10–40 (*Jackson and Redman, 2003*; *Delvendahl et al., 2015*).

To constrain the peak $Na^+$ conductance densities ($\bar{g}_{Na}$) we injected a 3 ms square current pulse in the somatic compartment and iteratively adjusted $\bar{g}_{Na}$ and $\bar{g}_K$. We varied both their peak conductance densities as well as the voltage-dependence of activation of $Na_V$ and $K_V$ channels by constraining the model AP to the experimentally recorded AP of the same neuron, with the aim to match the AP both in the *V-t* as well as the phase-plane dimensions recorded and simulated at 100 kHz (*Figure 7c*). To further constrain $\bar{g}_{Na}$ we compared the AP-evoked $[Na^+]$ with the experimental recordings using the ratiometric indicator SBFI, yielding a $\Delta[Na^+]_i$ of on average ~$1.5 \pm 0.2$ mM ($n =$ 5, imaged at 0.5 kHz; *Figure 8—figure supplement 2*). $Ca_V$ channels were incorporated based on previously published models (*Mainen and Sejnowski, 1996*) and $Ca_V$ channel conductance was separated in high- and low-voltage (T-type) activated channels and was varied between 2 and 4 pS µm$^{-2}$ in the AIS, 8 and 4 pS µm$^{-2}$ in the soma and ranged between 0.5 and 4 pS µm$^{-2}$ in the dendrites.

## Statistics and data availability

All statistical tests were done in GraphPad Prism 8 (version 8.1.2, GraphPad Software, Inc). Sample sizes for the pharmacological experiments were estimated based on the following assumptions: to observe a 50% block (based on *Bender and Trussell, 2009*) with 25% standard deviation (relative to mean) with a power of 0.8 and a type I error probability of 0.05, we would need a minimum of 4 paired recordings per treatment (PS Software version 3.1.6).

The cutoff significance level (*P*) was 0.05. Control peak $\Delta F/F$ values at the AIS in response to both subthreshold and AP signals were tested for normality. Since both data sets passed the D'Agostino and Pearson normality test, parametric tests were used to test all differences between peak OGB-1 $\Delta F/F$. To compare the spatial differences in signal amplitude we used one-way ANOVAs with multiple comparisons with Tukey correction for false positives. A linear regression was used to assess the synaptopodin and AIS marker length (Ankyrin G or ß4-spectrin) relationship. We used one-tailed ratio (compared log differences in the data set) paired *t*-test when analyzing all our toxin data. Differences between toxin and control give a measure of absolute reduction; differences between logarithms give a measure of relative reduction log toxin – log control = log (toxin/control). One tailed test was used on the premise that toxins reduce $Ca^{2+}$ signals. The exception was the effect of NCX for which we used a two-tailed ratio paired t-test. The OGB-5N peaks in response to subthreshold depolarizations passed the D'Agostino and Pearson normality test and to compare the effects of TTX and QX-314 on subthreshold $Ca^{2+}$ responses we used two-way-ANOVA with Sidak's correction for false positives. The following data sets passed the Shapiro-Wilk test for normality, so we compared the means using parametric tests: rise and decay times of OGB1 before and after $Ca^{2+}$ store release block, peak $\Delta F/F$ JPW3028, peak $\Delta F/F$ OGB-5N and slopes of $I_{Na}$, $I_{Ca\_opt}$ and $I_{Ca}$, the ratio of subthreshold peak to AP peak between sodium (SBFI) and calcium (OGB-5N) fluorescence. The following data did not pass the Shapiro-Wilk test for normality, so we compared the means using nonparametric tests: the slope AP peak between sodium (SBFI) and calcium (OGB-5N) fluorescence, the peak $I_{Na}$ and peak OGB-1 $\Delta F/F$ in HEK-293 cells. All data generated or analyzed are in the manuscript or supporting files. The source data files are provided for *Figures 1–7* and *Table 1*. The NEURON model morphology is available at NeuroMorpho.Org ID: NMO_75667, Neuron Name 2014-04-01_1 and the mod file used to model $Ca^{2+}$ diffusion and buffering is available at ModelDB, accession 125745 (https://senselab.med.yale.edu/ModelDB/), with adjustments described in 'Model simulations with a multicompartmental model'.

## Acknowledgements

The authors are indebted to Stefan Hallermann (Univ. Leipzig), Christian Lohmann (NIN–KNAW) and Christiaan de Kock (CNCR, VU) for critically reading earlier versions of the manuscript. We thank Huib Mansvelder (CNCR, VU) for sharing the human tissue. Part of the work was supported by the National Multiple Sclerosis Society (RG 4924A1/1) and the Netherlands Organization for Scientific Research (NWO), including an NWO Vici grant 865.17.003 and a Program Grant 16NEPH02 of the

Foundation for Fundamental Research on Matter (FOM) to MK OPA was supported by the Erasmus + Traineeship Program and the Graduate School of Systemic Neurosciences.

## Additional information

### Funding

| Funder | Grant reference number | Author |
|---|---|---|
| Nederlandse Organisatie voor Wetenschappelijk Onderzoek | 865.17.003 | Maarten HP Kole |
| Stichting voor Fundamenteel Onderzoek der Materie | 16NEPH02 | Maarten HP Kole |
| National Multiple Sclerosis Society | RG 4924A1/1 | Maarten HP Kole |

The funders had no role in study design, data collection and interpretation, or the decision to submit the work for publication.

### Author contributions

Naomi AK Hanemaaijer, Data curation, Software, Formal analysis, Validation, Investigation, Visualization, Methodology, Writing - review and editing; Marko A Popovic, Conceptualization, Software, Formal analysis, Supervision, Validation, Investigation, Visualization, Methodology, Writing - review and editing; Xante Wilders, Sara Grasman, Oriol Pavón Arocas, Investigation; Maarten HP Kole, Conceptualization, Formal analysis, Supervision, Funding acquisition, Validation, Visualization, Methodology, Writing - original draft, Project administration, Writing - review and editing

### Author ORCIDs

Naomi AK Hanemaaijer (iD) https://orcid.org/0000-0002-0329-5129
Oriol Pavón Arocas (iD) https://orcid.org/0000-0001-5822-8858
Maarten HP Kole (iD) https://orcid.org/0000-0002-3883-5682

### Ethics

Human subjects: Written informed consent was obtained from patients and all procedures on human tissue were performed with the approval of the Medical Ethical Committee of the Amsterdam UMC, location VuMC and in accordance with Dutch license procedures and the Declaration of Helsinki. All data were anonymized.
Animal experimentation: All animal experiments were performed in compliance with the European Communities Council Directive 2010/63/EU effective from 1 January 2013. They were evaluated and approved by the national CCD authority (license AVD8010020172426) and by the KNAW animal welfare and ethical guidelines and protocols (DEC NIN 14.49, DEC NIN 12.13, IvD NIN 17.21.01 and 17.21.03).

### Decision letter and Author response

Decision letter https://doi.org/10.7554/eLife.54566.sa1
Author response https://doi.org/10.7554/eLife.54566.sa2

## Additional files

### Supplementary files

• Transparent reporting form

### Data availability

All data generated or analyzed during this study are included in the manuscript and supporting files. Source data files have been provided for Figures 1 to 7 and Table 1.

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
