## [Decision Letter]

**Acceptance summary:**

The source of calcium entry at sites of action-potential initiation has been incompletely understood. In this study, quantitative electrophysiological, imaging, and computational approaches are used to demonstrate that voltage-gated sodium channels have a minor permeability to calcium that mediates a significant portion of depolarization-dependent calcium influx. This novel finding provides a comprehensive explanation of axonal calcium entry in mammalian cells.

**Decision letter after peer review:**

Thank you for submitting your article "Ca^2+^ entry through Na_V_ channels generates submillisecond axonal Ca^2+^ signaling" for consideration by *eLife*. Your article has been reviewed by four peer reviewers, including Merritt Maduke as the Reviewing Editor and Reviewer #1, and the evaluation has been overseen by Richard Aldrich as the Senior Editor. The following individuals involved in review of your submission have agreed to reveal their identity: Bernardo L Sabatini (Reviewer #3); James S Trimmer (Reviewer #4).

The reviewers have discussed the reviews with one another and the Reviewing Editor has drafted this decision to help you prepare a revised submission.

Summary:

This comprehensive study seeks to show that a source of calcium at the axon initial segment is permeation of Ca^2+^ through voltage gated Na channels. The quality and variety of quantitative electrophysiological, imaging and computational approaches that are brought to this problem is impressive. The question has been addressed previously in pyramidal cells and other neurons, and a general previous conclusion is that various calcium selective channels, Ca_V_s, are at play. However, in previous investigations, inhibition of Ca_V_s was only able to partially block influx, indicating an incomplete understanding. Here, the authors add to the story by concluding that the small calcium permeability of voltage-gated sodium channels (Na_V_s) to Ca mediates a significant portion of depolarization-dependent Ca influx in mammalian cells. This novel finding will be of wide interest. However, some key control experiments are essential to support the conclusions. In addition, a reframing of the Introduction and Discussion is essential to clarify the contribution of this report. The results are organized well and do a good job building the story. With the requested controls together with a reframed Introduction and Discussion, this paper should be a very enjoyable read.

Essential revisions:

1) A major concern deals with the use of Ca_V_ blockers in Figure 3. There are no positive controls for any of the 7 compounds. Yet the reader needs to take this array of negative results as major support for the work's conclusions. For the peptides, SNX, conotoxin, there is no indication that the drugs were applied using solutions in which protein binding was pre-inhibited with BSA or cytochrome C. While at ~1 μm it likely won't matter much; however, it is a standard step. With puffing of the drug, the final concentration would be expected to be ~10% of the pipette concentration, again calling up the need for a positive control.

Figure 3 also shows that there are small reductions with TTA, Ni, Isradepine, Nimodipine. If you add those up, and take into account the point above, then perhaps a significant amount of the Ca enters through Ca_V_s, and thus the relative contribute of Na_V_s and Ca_V_s remains uncertain. Perhaps combinations of blockers could give a more accurate picture.

2) Figure 5 is a fascinating and creative experiment that potentially provides a potent argument for the Na_V_ source of Ca. However, there remain concerns of whether the voltage signals before and after the TTX+voltage increase manipulation are really reflective of equivalent membrane potentials. How linear is the JPW signal? And how stable? How does the signal linearity change when the there is a large overshoot? If the transients are not an accurate reflection of the relative magnitude of AIS voltage change then the experiment is less compelling. Since the author are masters of bleb recording, why not use this to get a more direct *V*_m_ assessment than V imaging?

3) Electrophysiological analysis. The authors should analyze AP properties recorded in current-clamp data. The APs half-width and after-depolarization should be characterized. Each independent pharmacological manipulation should be compared to the control condition and the EGTA condition, which should provide constraints on the effect of calcium influx on the AP waveform. Of particular interest is whether calcium influx through voltage-gated calcium or sodium channels has particular importance for calcium-activated potassium channels, which may be linked through nanodomain signaling. An example of this effect might be seen in the *V*_m_ traces of Figure 3A and C (the Aps look quite different with T-type block), but it is not discussed. Furthermore, slight differences in AP waveform can have profound effects on calcium influx through voltage-gated calcium channels. These effects should be ruled out when relevant. This analysis should be accompanied by discussion of Na_V_-mediated calcium influx's function in the AIS. For example, if the small calcium signal produced by T-Type channels is primarily responsible for shaping the afterdepolarization, then the authors should speculate on the function of the Na_V_-mediated calcium signal.

The results of these analyses are relevant for an overstated claim in the first paragraph of the Discussion. The authors write "[sodium channels] generate local and rapid Ca-dependent biochemical signaling". No experiment or analysis was done to directly evaluate this claim. Ca-dependent biochemical signaling is not addressed at all. This could be tested by examining the opening of Ca-activated K channels by Na_V_-dependent Ca influx. The "local" part may also be problematic. If local means "in the same subcellular domain as the open Ca-permeable channel" then this is trivially true. If it means something about nanodomain signaling, this has not been examined (but might potential be with analysis of Ca-activated K channels).

4) It would also be of interest to know if Na_V_-mediated calcium influx impacts cellular firing patterns, such as spike accommodation. It is not clear from the manuscript whether this analysis could be performed without additional experimentation – if so, it is certainly worth analyzing the data in hand to discover such potential effects.

5) Ca release is blocked by applying 200 μm ryanodine and 5 mg/ml heparin to the intracellular solution, resulting in a nearly 50% reduction of the Ca signal. This is ~ 20 times more ryanodine than is usually bath applied to block store release. How is this concentration justified and could it possibly have any off-target effects at that dose?

6) What efforts were made to confirm that Ca signals are not saturating? This would impact kinetics or additive effects greatly.

7) Writing. The authors attempt to make two claims that are misleading and unnecessary for the impact of the manuscript. First, they claim that Na_V_-mediated calcium signals are a novel finding. They do this in the Abstract and in the first paragraph of the Discussion. However, they cite the multiple papers that have reported this phenomenon in squid axons. I suggest that the authors couch the novelty of their result in this literature, and instead claim that this source is novel in mammalian cells, as is done in the third paragraph of the Discussion. This would clarify the contribution of this report and deflect any unnecessary criticism about the accuracy of its claims.

Next, they write, "the source of Ca transients at the axon initial segment and nodes of Ranvier.… is less understood". I find that this does not accurately represent the state of the literature. There is significant knowledge related to calcium signaling in axons, including their effect on AP waveform via their interaction with calcium-activated potassium channels and the role of calcium channels in generating cell-type specific firing patterns. Accompanying these effects are multiple papers that use pharmacological compounds to identify the source of the calcium influx, all of which have failed to completely block axonal calcium signals. This point is key to the novelty and impact of Popovic et al.'s manuscript. Instead of arguing ambiguously that axonal calcium signaling is poorly understood, we recommend that Popovic et al. explicitly discuss how the inability to completely block axonal calcium signals, together with the historical observation of a TTX-sensitive calcium current, demand a revision of our current model for mammalian axonal calcium signaling. This framework would be more accurate, informative, and erudite. This consideration should be propagated throughout the entire manuscript.

8) In the spirit of rigor and transparency contributing to reproducibility, the authors need to provide more detail on toxin and blockers used. This should include both the source and the source and catalog numbers, similar to what they included for amplifiers, objectives, etc. Attempts to reproduce and/or extend this work rely of using the correct reagents and there is little to no detail on exactly what was used. This also holds for the antibodies used for the immunolabeling. The authors state they used these antibodies: Synaptopodin (rabbit; 1:500; Sigma-Aldrich Chemie); Ankyrin G (mouse; 1:100; Neuromab) and ßIV-spectrin (mouse; 1:250; Neuromab). However, from their websites, Sigma sells six different rabbit anti-synaptopodin antibodies, and NeuroMab distributes four different anti-Ankyrin G antibodies, and two different anti- ßIV-spectrin antibodies, all six in two different forms (pure and tissue culture supernatant). In looking at the website info the different antibodies against these targets have very different characteristics, supporting the importance of defining which specific ones were used. We suggest the authors include catalog numbers and RRID numbers for unambiguous identification of the reagents used. *eLife* suggests/requires a key resources table to promote rigor and transparency. This would be an effective way to include these details.

---

## [Author Response]

Essential revisions:1) A major concern deals with the use of Ca_V_ blockers in Figure 3. There are no positive controls for any of the 7 compounds. Yet the reader needs to take this array of negative results as major support for the work's conclusions. For the peptides, SNX, conotoxin, there is no indication that the drugs were applied using solutions in which protein binding was pre-inhibited with BSA or cytochrome C. While at ~1 μm it likely won't matter much; however, it is a standard step. With puffing of the drug, the final concentration would be expected to be ~10% of the pipette concentration, again calling up the need for a positive control.

We have now performed new experiments to examine the efficacy of drug application. More specifically, we repeated the ω-conotoxin MVIIC (a N-, P-/Q-type Ca_V_ channel blocker) puffing experiment with addition of 0.1% BSA in the ASCF, to reduce nonspecific binding of the drug. The effect of the drug was not altered; ω-conotoxin MVIIC led to a non-significant trend to reduce the AP-evoked Ca^2+^ influx in the AIS by ~7% (*n* = 5, see Results and Figure 3B). In contrast, as a positive control experiment we imaged the Ca^2+^ in collateral boutons from the same axons, where the pressure application of the toxin from the same pipette reduced the AP-induced Ca^2+^ influx by ~90% (see Results, *P* = 0.021, *n* = 3; Figure 3B).

Figure 3 also shows that there are small reductions with TTA, Ni, Isradepine, Nimodipine. If you add those up, and take into account the point above, then perhaps a significant amount of the Ca enters through Ca_V_s, and thus the relative contribute of Na_V_s and Ca_V_s remains uncertain. Perhaps combinations of blockers could give a more accurate picture.

The channels that significantly contributed to AP-evoked Ca^2+^ influx were T- and L-type Ca_V_ channels. To address whether their block is additive and thereby accounts for a larger fraction of the Ca^2+^ influx we performed new experiments combining the T- and L-type blockers TTA-P2 and isradipine, respectively. We did not combine all four blockers, since they have shared targets and combining blockers with the same targets is in essence similar to increasing the blocker concentration, which may lead to off-target effects. TTA-P2 and isradipine were selected since they individually showed the most robust effect (least variance). The observed block was a 27.2% reduction in the AP-evoked peak in ∆F/F (relative to control, *P* = 0.0071, *n* = 5; Figure 3B). This block was less in comparison to the linear sum of block 33.6% (TTA-P2, 18.7% + Isradipine, 14.9%) and further underscores the idea that voltage-gated Ca^2+^ channels at the AIS are not the only source for Ca^2+^ influx.

2) Figure 5 is a fascinating and creative experiment that potentially provides a potent argument for the Na_V_ source of Ca. However, there remain concerns of whether the voltage signals before and after the TTX+voltage increase manipulation are really reflective of equivalent membrane potentials. How linear is the JPW signal? And how stable? How does the signal linearity change when the there is a large overshoot? If the transients are not an accurate reflection of the relative magnitude of AIS voltage change then the experiment is less compelling. Since the author are masters of bleb recording, why not use this to get a more direct V_m_ assessment than V imaging?

We are pleased to read the reviewers appreciated the experimental design. Regarding the possibility to record in whole-cell patch-clamp configuration from axon blebs; this approach has its advantages and disadvantages. While it has excellent signal-to-noise for voltage recordings, in transected axon blebs intracellular Ca^2+^ is increased and plays a role in the cytoskeletal organization for resealing and regrowth (Bradke et al. Nat Rev Neurosci 13, 2012). Since the focus of the present study is the dynamics of intracellular Ca^2+^ under physiological conditions we decided that electrical recording (or imaging) from blebs will have confounding effects.

Using a high-speed imaging experiments strategy, we can record from unperturbed axons. The synthetic dye JPW has a very fast activation time constant (~11 µs) and is linear over a large voltage range (–125 to 125 mV, Ehrenberg et al., Biophys J 51, 1987) making it the voltage-sensitive dye of choice to track fast action potentials in neurons and in particular in axons (Cohen et al. Cell 180, 2020; Popovic et al. The Journal of Physiology 589, 2011). The reviewers raised two concerns regarding voltage imaging in this experiment: the stability and the linearity of the voltage dye. To address the first concern, we measured the baseline fluorescence of all recordings shown in Figure 5 and did not observe a change in baseline fluorescence between conditions (data not shown, *P* = 0.452, ordinary one-way ANOVA, *n* = 9), confirming that the dye is stable over the time period in which the experiments were performed. To address the linearity of the dye: we agree that linearity had not been examined over the larger voltage range as was used here (from resting membrane potential to +170 mV). In the revised manuscript we addressed this in HEK293 cells, in which a reliable and stable voltage-clamp can be maintained. We performed experiments in which JPW3028 fluorescence was imaged while clamping the cell across the range of potentials used in the AP-clamp experiment in Figure 5. The results show that the relationship between JPW ∆F/F and membrane potential is linear between –75 up to +175 mV, with Pearson’s correlation coefficient r > 0.99 in all cells recorded (see Figure 5—figure supplement 1).

3) Electrophysiological analysis. The authors should analyze AP properties recorded in current-clamp data. The APs half-width and after-depolarization should be characterized. Each independent pharmacological manipulation should be compared to the control condition and the EGTA condition, which should provide constraints on the effect of calcium influx on the AP waveform. Of particular interest is whether calcium influx through voltage-gated calcium or sodium channels has particular importance for calcium-activated potassium channels, which may be linked through nanodomain signaling. An example of this effect might be seen in the V_m_ traces of Figure 3A and C (the Aps look quite different with T-type block), but it is not discussed. Furthermore, slight differences in AP waveform can have profound effects on calcium influx through voltage-gated calcium channels. These effects should be ruled out when relevant. This analysis should be accompanied by discussion of Na_V_-mediated calcium influx's function in the AIS. For example, if the small calcium signal produced by T-Type channels is primarily responsible for shaping the afterdepolarization, then the authors should speculate on the function of the Na_V_-mediated calcium signal.

We agree with the reviewers the AP changes may be informative and add to our understanding of the function of Na_V_ mediated Ca^2+^ influx. In the revised version we now show the analysis of individual APs recorded from the soma in Figure 9 and Table 1. Two conditions with the largest effect on Ca^2+^ influx were compared: the combined block of T- and L-type Ca_V_ channels and preventing all Ca^2+^ influx by lowering extracellular Ca^2+^ concentration by bath application of EGTA (Figure 3). Blocking T- and L-type Ca^2+^ channels reduced the afterdepolarization while leaving the AP half-width unaffected (ADP, mean difference = –2.9 mV, *P* = 0.042, FWHM, mean difference = 0.05 ms, *P* = 0.17, *n* = 5). In contrast, lowering extracellular Ca^2+^ by EGTA increased the AP half-width and afterhyperpolarization (AP half-width, mean difference = 0.29 ms, *P* = 0.018, AHP, mean difference = +19.1 mV *P* = 0.048, *n* = 4). These data suggest that the Ca^2+^ sources affect somatic AP shape differentially: the T- and L-type channels play a role in the delayed afterdepolarization while in the EGTA condition, also the submillisecond-rapid Na_V_-mediated Ca^2+^ entry is abolished, which potentially provides a route to activate BK channels and shorten the AP half-width. While we have not directly examined BK_Ca_ channels, these data are consistent with the work of Bock and Stuart, 2016, and Yu et al., 2010, showing that Iberiotoxin (IbTx)-mediated block of BK_Ca_ channels increases the AP width duration at the soma of L5 neurons and AIS. Interestingly, local IbTx application at the first node of Ranvier also increases the half-width duration where Ca^2+^ uncaging shortens the half-width (Roshchin et al., 2018). We have now included these ideas in the Discussion.

The results of these analyses are relevant for an overstated claim in the first paragraph of the Discussion. The authors write "[sodium channels] generate local and rapid Ca-dependent biochemical signaling". No experiment or analysis was done to directly evaluate this claim. Ca-dependent biochemical signaling is not addressed at all. This could be tested by examining the opening of Ca-activated K channels by Na_V_-dependent Ca influx. The "local" part may also be problematic. If local means "in the same subcellular domain as the open Ca-permeable channel" then this is trivially true. If it means something about nanodomain signaling, this has not been examined (but might potential be with analysis of Ca-activated K channels).

With local we were referring to the fact that the AIS and nodes show hot spots for Ca^2+^ influx (Figure 1). However, we agree with the reviewers we do not have evidence for biochemical signaling and removed such claims from the manuscript. Our data indicate that AP width is Ca^2+^ dependent but T- and L-type Ca^2+^ channel independent, leaving open the possibility that rapid Na_V_-channel mediated Ca^2+^ influx is a trigger for BK channel opening (Figure 9 and Table 1). These findings are well in agreement with the work of Roshchin et al., 2018, showing that AIS and node of Ranvier Ca^2+^ activates BK channel-mediated repolarization of the AP in the L5 pyramidal neuron. Nevertheless, to provide direct evidence that it is the Na_V_-mediated Ca^2+^ entry that activates BK channels would require knocking down the molecular mechanism mediating the transfer of Ca^2+^ through the pore, performing direct voltage-clamp of BK currents, etc. Such additional experiments go beyond the aim of the present study to identify the sources of axonal Ca^2+^ entry.

4) It would also be of interest to know if Na_V_-mediated calcium influx impacts cellular firing patterns, such as spike accommodation. It is not clear from the manuscript whether this analysis could be performed without additional experimentation – if so, it is certainly worth analyzing the data in hand to discover such potential effects.

This point has been answered in more detail above in point #3.

5) Ca release is blocked by applying 200 μm ryanodine and 5 mg/ml heparin to the intracellular solution, resulting in a nearly 50% reduction of the Ca signal. This is ~ 20 times more ryanodine than is usually bath applied to block store release. How is this concentration justified and could it possibly have any off-target effects at that dose?

The ryanodine concentration we used is indeed 20 times higher compared to the reported low affinity *K*_d_ of RyRs (10 µM, Sutko et al. Pharmacol Rev. 49, 1997). This is a similar concentration as applied previously in combination with high speed imaging (Jaafari, De Waard and Canepari, 2014). These authors found no impact on peak ∆F/F evoked by single AP-induced Ca^2+^ in dendrites with blockers before and after intracellular application of ryanodine receptor block. If off-target effects are present they are thus not detectable in dendrites. That we find a change in the AIS is consistent with the large ER organelle in these pyramidal neurons (Figure 2).

6) What efforts were made to confirm that Ca signals are not saturating? This would impact kinetics or additive effects greatly.

In some of our imaging experiments we also recorded the fluorescence in response to multiple APs for both OGB-1 and OGB-5N. We observed, however, that the peak ∆F/F in response to a single AP is well below dye saturation (OGB-1, 25 ± 2.7 %, *n* = 8 and OGB-5N, saturation not observed with 3 spikes, *n* = 3). We are therefore confident that the Ca^2+^ fluorescence we quantified in this paper (subthreshold stimuli and a single AP) are in the ∆F/F range from dye saturation.

7) Writing. The authors attempt to make two claims that are misleading and unnecessary for the impact of the manuscript. First, they claim that Na_V_-mediated calcium signals are a novel finding. They do this in the Abstract and in the first paragraph of the Discussion. However, they cite the multiple papers that have reported this phenomenon in squid axons. I suggest that the authors couch the novelty of their result in this literature, and instead claim that this source is novel in mammalian cells, as is done in the third paragraph of the Discussion. This would clarify the contribution of this report and deflect any unnecessary criticism about the accuracy of its claims.

We have now revised these statements, removed primary claims and thoroughly rewrote the Abstract, Introduction and parts of the Discussion. The first line in the Discussion now reads “We identified Ca^2+^ permeation through Na_V_ channels as a source for activity-dependent Ca^2+^ entry in mammalian axons”.

Next, they write, "the source of Ca transients at the axon initial segment and nodes of Ranvier.… is less understood". I find that this does not accurately represent the state of the literature. There is significant knowledge related to calcium signaling in axons, including their effect on AP waveform via their interaction with calcium-activated potassium channels and the role of calcium channels in generating cell-type specific firing patterns. Accompanying these effects are multiple papers that use pharmacological compounds to identify the source of the calcium influx, all of which have failed to completely block axonal calcium signals. This point is key to the novelty and impact of Popovic et al.'s manuscript. Instead of arguing ambiguously that axonal calcium signaling is poorly understood, we recommend that Popovic et al. explicitly discuss how the inability to completely block axonal calcium signals, together with the historical observation of a TTX-sensitive calcium current, demand a revision of our current model for mammalian axonal calcium signaling. This framework would be more accurate, informative, and erudite. This consideration should be propagated throughout the entire manuscript.

We thank the reviewer for the suggestion about positioning the research findings. As indicated above, we have followed the suggestions and revised the manuscript. More specifically, we toned down novelty claims and now more clearly indicate how Ca^2+^ influx through Na_V_ channels confirms and extends previous studies reporting TTX-dependent and Ca_V_ -dependent Ca^2+^ influx in axons.

8) In the spirit of rigor and transparency contributing to reproducibility, the authors need to provide more detail on toxin and blockers used. This should include both the source and the source and catalog numbers, similar to what they included for amplifiers, objectives, etc. Attempts to reproduce and/or extend this work rely of using the correct reagents and there is little to no detail on exactly what was used. This also holds for the antibodies used for the immunolabeling. The authors state they used these antibodies: Synaptopodin (rabbit; 1:500; Σ-Aldrich Chemie); Ankyrin G (mouse; 1:100; Neuromab) and ßIV-spectrin (mouse; 1:250; Neuromab). However, from their websites, Σ sells six different rabbit anti-synaptopodin antibodies, and NeuroMab distributes four different anti-Ankyrin G antibodies, and two different anti- ßIV-spectrin antibodies, all six in two different forms (pure and tissue culture supernatant). In looking at the website info the different antibodies against these targets have very different characteristics, supporting the importance of defining which specific ones were used. We suggest the authors include catalog numbers and RRID numbers for unambiguous identification of the reagents used. eLife suggests/requires a key resources table to promote rigor and transparency. This would be an effective way to include these details.

We have included a completed Key Resources table in the Materials and methods section.